# Lightweight and drift-free magnetically actuated millirobots via asymmetric laser-induced graphene

Yun Chen [1], Yuanhui Guo[1], Bin Xie[1], Fujun Jin [2], Li Ma[1], Hao Zhang[1], Yihao Li[3], Xin Chen [1] ✉, Maoxiang Hou[1], Jian Gao[1], Huilong Liu[1], Yu-Jing Lu [2] ✉, Ching-Ping Wong [4] & Ni Zhao [3] ✉

Millirobots must have low cost, efficient locomotion, and the ability to track target trajectories precisely if they are to be widely deployed. With current materials and fabrication methods, achieving all of these features in one millirobot remains difficult. We develop a series of graphene-based helical millirobots by introducing asymmetric light pattern distortion to a laser-induced polymer-to-graphene conversion process; this distortion resulted in the spontaneous twisting and peeling off of graphene sheets from the polymer substrate. The lightweight nature of graphene in combine with the laser-induced porous microstructure provides a millirobot scaffold with a low density and high surface hydrophobicity. Magnetically driven nickel-coated graphene-based helical millirobots with rapid locomotion, excellent trajectory tracking, and precise drug delivery ability were fabricated from the scaffold. Importantly, such high-performance millirobots are fabricated at a speed of 77 scaffolds per second, demonstrating their potential in high-throughput and large-scale production. By using drug delivery for gastric cancer treatment as an example, we demonstrate the advantages of the graphene-based helical millirobots in terms of their long-distance locomotion and drug transport in a physiological environment. This study demonstrates the potential of the graphene-based helical millirobots to meet performance, versatility, scalability, and cost-effectiveness requirements simultaneously.

Millirobots are machines with submillimeter dimensions that are capable of moving autonomously to perform specific tasks in microscopic confined spaces[1-3], such as microfluidic channels, biochips, and even the blood vessels of living organisms. In the past decade, proof-of-concept millirobots for drug or gene delivery[4-8], pollutant cleaning[9-12], and sensing[13-15] applications have been reported. However, the steering and locomotion abilities of millirobots, in particular

their movement velocity and lateral drift[16], must still be improved to meet reliability and controllability requirements for their practical use.

Many millirobots have a helical configuration[17] because this configuration can support both self-propulsion (such as chemical propulsion[18]) and external-stimulus-based propulsion when the stimulus originates from a magnetic field[19,20], an optical field[21-23], or an acoustic field[24-26]. Several processing methods including three-

[1]State Key Laboratory of Precision Electronic Manufacturing Technology and Equipment, School of Electromechanical Engineering, Guangdong University of Technology, Guangzhou 510006, PR China. [2]Institute of Natural Medicine and Green Chemistry, School of Biomedical and Pharmaceutical Sciences, Guangdong University of Technology, Guangzhou 510006, PR China. [3]Department of Electronic Engineering, The Chinese University of Hong Kong, Shatin, Hong Kong, China. [4]School of Materials Science and Engineering, Georgia Institute of Technology, Atlanta, GA 30332, USA. ✉e-mail: chenx@gdut.edu.cn; luyj@gdut.edu.cn; nzhao@ee.cuhk.edu.hk

dimensional direct laser lithography[27–29], glancing angle deposition[30,31], biotemplate[32,33], laser ablation[34], and origami-based self-scrolling[35–37] have been developed to create helical microstructures made from photoresists, hydrogels, or metals for millirobot fabrication. These methods' relatively low throughput limits the scalability of millirobots; therefore, these millirobots are relatively expensive. Furthermore, because the materials used to fabricate millirobots are considerably denser than the liquids that these devices operate in, the millirobots might gradually sink during movement, thereby resulting in a major drift of the devices from their desired trajectory and a speed reduction[16,27,28,34,38–41]. Accordingly, additional control units are required to offset the effects of gravity, and these units increase the complexity and price of the entire millirobot system.

We developed an unconventional laser-induced graphene (LIG) process that enables the high-throughput fabrication of millirobots with high speed and nearly-drift-free orientation capabilities. Through beam shaping and defocusing, triangular laser spots are generated and aligned along the laser scanning direction to create a sloped intensity profile for local processing; when such a light pattern is used to convert a polymer surface region into graphene, uneven gas fluxes are induced, and the forces at different edges of the graphene sheet are thus unbalanced, which causes the sheet to twist into a helical configuration as it peels off the polymer substrate. More importantly, the lightweight nature of graphene in combination with the highly porous microstructure created by the laser process results in a millirobot scaffold that has low density and high surface hydrophobicity; the resultant millirobots thus propel themselves while being fully suspended rather than exhibiting on-ground-like movement. In this study, magnetically driven nickel-coated graphene-based helical (GH) millirobots were found to exhibit almost zero lateral drift while achieving a high swimming velocity of 2.64 body lengths per second (forward velocity of 3109 µm/s). By systematically varying the laser processing conditions, we demonstrated that our fabrication method offers high throughput (77 millirobot scaffolds per second) and is versatile; various geometrical parameters of the millirobot can be tuned, including its body length, its sheet width, and the chirality, diameter, angle, and pitch of the helical structure. Finally, we used therapeutic drug delivery for gastric cancer treatment as an example to illustrate the

advantages of the developed GH millirobots for long-distance locomotion and drug transport in a physiological environment.

## Results

### Processing and characterization of porous helical LIG sheets

To ensure that our millirobots would be light, thus achieving the desired balance between gravity and buoyancy for precise propulsion (see the Supplementary Information for a detailed discussion), we designed and fabricated a porous GH millirobot. An ultraviolet (UV) laser system with a flat-field scanning galvanometer was developed to create helical graphene sheets from a polyimide (PI) substrate; a schematic of the equipment is shown in Fig. 1a. The flat-field scanning galvanometer was a biaxial scanning system that deflected the laser beam in the X-direction and Y-direction to form a two-dimensional scanning area, which was defined as the working plane of the laser. A set of F-θ focusing lenses was placed under the galvanometer and aligned along its optical axis for beam shaping. In a commonly used setting, the deflected laser beam is focused to a circular spot on the working plane, as illustrated in Supplementary Fig. S1a. In our fabrication platform, however, we intentionally tilted one of the F-θ lenses to shift the laser beam away from the original focusing surface; this shift induced asymmetric distortion of the laser spot on the working plane. By appropriately adjusting the distance between the working plane and the focal plane (i.e., the defocusing distance), we obtained a laser spot with a triangular shape on the working plane, as illustrated in Fig. 1a and Supplementary Fig. S1b. The predicted geometries of laser spots under normal and distorted conditions were experimentally verified by obtaining scanning electron microscopy (SEM) images of the laser-created spots on a substrate (Fig. 1b, c). Although the defocusing method was a convenient approach for achieving the triangular laser spot, there is a limit on the defocusing distance (typically 10 mm); beyond a distance of 10 mm, the beam quality factor of the laser spot deteriorated considerably, which rendered the laser incapable of reliable and efficient processing[42,43].

The triangular laser spot was leveraged to produce porous and helical graphene sheets from a PI thin film. PI was selected as the starting material because it contains not only carbon but also hydrogen, oxygen, and nitrogen. Consequently, during the formation of LIG, various gas species, such as $CO_2$, $CO$, $H_2O$, and $NO_2$, were

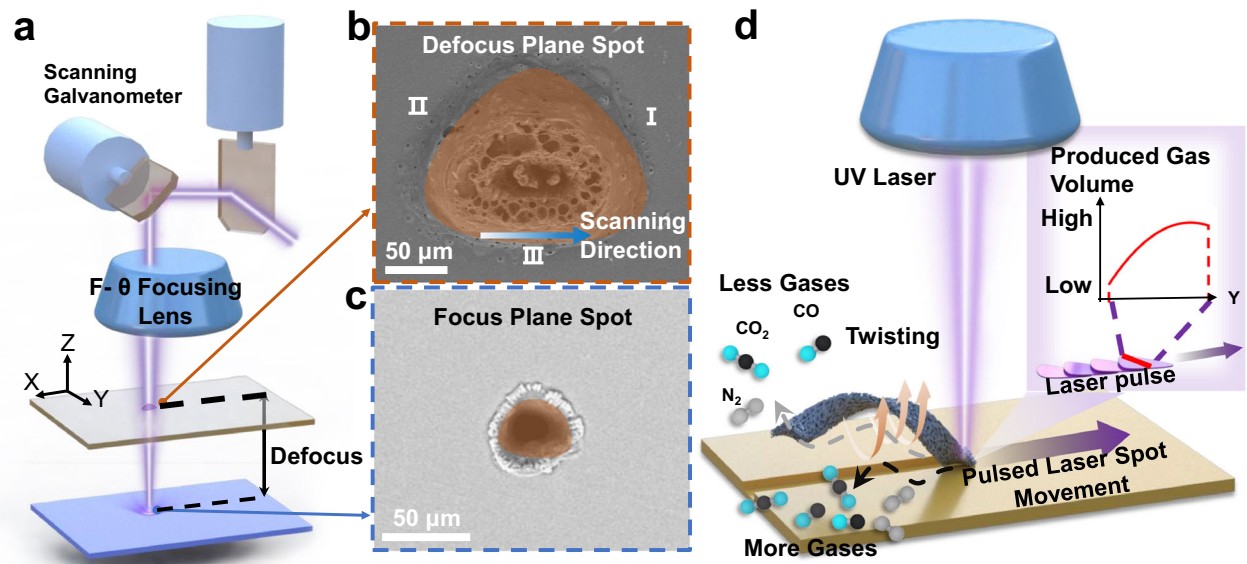

**Fig. 1 | Design of and process for producing porous helical LIG sheets.**
**a** Modulation for creating laser spots of different shapes at various distances from the focus plane. **b** Laser spot in the shape of a rounded triangle on the defocus plane. **c** Round laser spot on the focus plane. **d** Schematic of the fabrication of a helical LIG sheet by using the laser spot on the defocus plane.

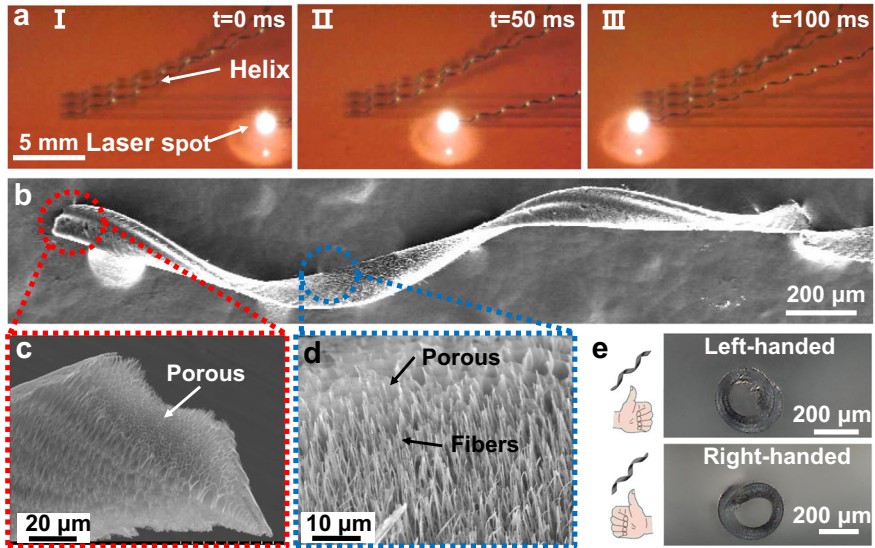

**Fig. 2 | Processing and characterization of porous helical LIG sheets. a** High-speed camera images of the formation of porous helical LIG sheets. **b** SEM image of a helical LIG sheet. **c** Porous cross section of the helical LIG sheet and **d** tree-like fibers at the sheet's bottom surface. **e** Right- and left-handed porous helical LIG sheets processed by selecting different laser scanning directions. These experiments were repeated for more than 10 different batches and yield similar results.

produced, which resulted in the creation of a substantial number of micropores and nanopores[44]. Increasing the laser power accelerated the production and release of the aforementioned gases and consequently enhanced the porosity of the LIG[45]. When the laser power exceeded 39 J/cm², the rate of gas production far exceeded the rate of gas release, thereby resulting in the accumulation of a considerable quantity of gas that eventually broke the weak connection between the top LIG sheet and the PI substrate. This disconnection meant that the LIG sheet peeled off the PI substrate. The formation of the graphene phase was confirmed through Raman spectroscopy (Supplementary Fig. S2a). Because the gas production was induced by the laser, the geometry of the laser spot influenced the final configuration of the LIG sheet. For instance, when the triangular laser spot was asymmetric along the laser scanning direction−for example, the laser scanning direction aligned with one of the edges of the triangle (e.g., edge III in Fig. 1b)−a considerably higher quantity of gas formed at the bottom edge of the LIG sheet than at its top edge because of more intense laser exposure at the bottom edge. This uneven gas formation created a torque force that deformed and twisted the LIG sheet. When torque force formation continued along the scanning line, the sheet was transformed into a helix, as illustrated in Fig. 1d. By contrast, when the laser spot was circular, the gas formation along the scanning line was uniform, which resulted in a bent LIG sheet (Supplementary Fig. S3).

The PI-to-LIG conversion process created a microstructure in the LIG sheets that favored the sheets' application as a millirobot scaffold. First, the high-flux gas production during graphenization created abundant micropores and nanopores throughout the sheets, as revealed by the cross-sectional SEM image displayed in Fig. 2c. This high porosity not only resulted in the sheets being less dense than they would have been without the pores but also rendered their surface hydrophobic. For instance, a laser-processed LIG sheet was discovered to modulate a contact angle within the range of 94°−131° by adjusting laser power levels (Supplementary Fig. S2b, S2c). Furthermore, the twisting and peeling off of the LIG sheet from the substrate stretched the microstructure at the sheet's bottom surface, leaving behind tree-like fibers (Fig. 2d). Pathways for capillary emptying[46] formed between the fibers, and these pathways further enhanced the hydrophobicity of the helical LIG sheets. Overall, the hydrophobic microstructure allows for the efficient trapping of air inside the pores when the helical LIG

sheet is immersed in a liquid, with this trapped air conferring favorable buoyancy to the sheet; moreover, the hydrophobicity of the millirobot's surface can also lead to a greater step-out frequency and higher swimming velocity[28].

Importantly, the fabrication of the helical LIG sheets was rapid and well-controlled. As displayed in Fig. 2a, only approximately 100 ms was required to produce a helical LIG sheet with a length of 12 mm (Supplementary Video 1); 77 helical graphene scaffolds with a typical length of 1.5 mm could thus be fabricated within only 1 s. The chirality of the helical structure can be set to right- or left-handed through the selection of the laser scanning direction (Fig. 2e and Supplementary Figs. S4 and S5). In addition, the geometrical parameters of the helical LIG sheets−including their width, helix diameter, pitch, and body length−can be precisely controlled by adjusting the laser power and scanning conditions. The smallest millirobot fabricated in this study had a sheet width, helix diameter, and helix pitch of approximately $86 \pm 4$, $167 \pm 4$, and $986 \pm 6$ μm, respectively (thus, the minimal length of a millirobot containing at least one full helix pitch is approximately 1 mm). At sizes greater than these minimal sizes, the dimensions of the millirobots can be tuned by varying the laser spot size, scanning speed, and scanning length, with the tuning resolutions for the sheet width, helix diameter, pitch, and length being 13, 47, 18, and 10 μm, respectively (see Supplementary Figs. S5 and S6 and the corresponding discussions for details). The density and surface wettability of the helical LIG sheets can be tuned by adjusting the laser power and scanning speed. Therefore, the suspension and motion characteristics of the millirobots can be decided in the fabrication stage.

## Manipulation and characterization of GH millirobots

To create magnetic-field-controllable GH millirobots from the fabricated helical LIG sheets, we sputtered a thin layer of nickel on the sheets and then conducted standard magnetization treatment along the radial direction of the helix. The results of SEM imaging and energy-dispersive X-ray spectroscopy (EDS) mapping (Supplementary Figs. S7 and S8) indicated that nickel was mostly distributed on the surface of the LIG sheets and that the lightweight and porous properties of the bulk LIG sheets were retained. Manipulation of the GH millirobots involved applying a rotating magnetic field **B** generated by a three-dimensional Helmholtz coil (Supplementary Fig. S9). This field produced a torque **M** with a rotational frequency $f$ on the GH

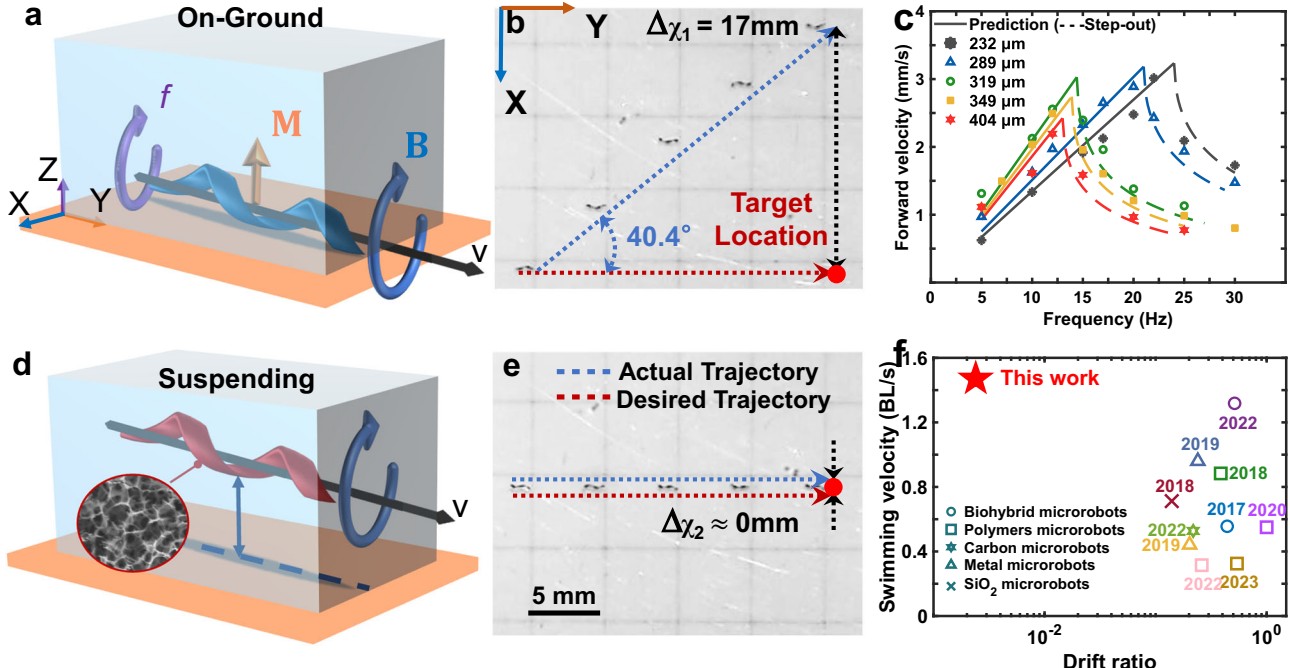

**Fig. 3 | Schematic of GH millirobots with different states and characterizations when suspended. a** Schematic of the high-density, on-ground-like GH millirobot. The torque **M** applied to this GH millirobot was produced using a uniformly strong rotating magnetic field **B** under a rotational frequency *f*. **b** Time-lapse diagram of the motion of the high-density GH millirobot (on-ground GH millirobot). Scale bar: 5 mm. **c** Velocity of low-density GH millirobots with helix diameters of 232, 289, 319, 349, and 404 μm at various rotational frequencies. The process used to obtain the prediction curve is detailed in Supplementary Note S1. **d** Schematic of the low-density fully suspended GH millirobot. **e** Time-lapse diagram of the motion of the low-density GH millirobot (suspending GH-millirobot). Scale bar: 5 mm. **f** Comparison of the swimming velocity [unit: body lengths per second (BL/s)] and drift ratio of magnetically actuated helical millirobots[3,16,28–30,40,48–51]. The details are shown in Supplementary Table S1.

millirobots; the torque initiated their rotation, and the helical configuration's motion was transformed into translational propulsion. Through the alteration of the rotational direction and magnitude of the magnetic field, the GH millirobots could be directionally steered. Unless otherwise specified, the motion of the GH millirobots was measured under a 12 mT magnetic field.

We first examined the effect of the fabricated GH millirobots' density on the accuracy of their trajectory. By adjusting the laser processing parameters, we fabricated two types of GH millirobots: a high-density (1.42 g/cm³) millirobot, which quickly sank in a liquid environment and exhibited on-ground-like conventional motion (Fig. 3a, the processing details are shown in Supplementary Fig. S10), and a low-density (1.023 g/cm³) millirobot, which was fully suspended in a liquid environment during propulsion (Fig. 3d). Notably, the low-density GH millirobot is the lightest magnetically actuated millirobot reported to date. As illustrated in Fig. 3b, e, both millirobots were expected to complete a 20 mm-long linear trajectory along the y-axis while being driven by the same rotating magnetic field. However, the high-density millirobot exhibited a pronounced lateral drift (i.e., drift along the x-axis) during propulsion (Supplementary Video 2). Consequently, the final trajectory of this millirobot deviated from the target trajectory by 40.4° (lateral drift Δχ of 17 mm; Fig. 3b). By contrast, the low-density millirobot exhibited almost zero lateral drift (Fig. 3e), demonstrating a remarkable ability in tracking the designated trajectories. Helical LIG sheets with different chirality have the same material properties and microstructure. Accordingly, millirobots made from helical LIG sheets with differing chirality have the same density but rotate in opposite directions under the same magnetic field, thereby resulting in them having opposite-direction movement, as shown in Supplementary Fig. S11 and Supplementary Video 3.

Subsequently, we evaluated the motion characteristics of the fabricated GH millirobots. The rotational frequencies of the applied magnetic field were comparable to those of other magnetic-field-driven millirobots. Under a high rotational frequency of 24 Hz, the lightest helical GH millirobot achieved a high forward velocity of −3109 μm/s, which was approximately equal to 2.64 body lengths per second (1 body length ≈ 1200 μm), in deionized water. For an equitable comparison of the propulsion speeds among different millirobots, we calculate the swimming velocity[47], defined as the number of body lengths advanced per second, for our millirobots, and compare the value with those reported in the literature[3,16,28–30,40,48–51]. As shown in Fig. 3f, the lightest GH millirobot exhibited a high swimming velocity of 1.43 body lengths per second when the rotational frequency was 10 Hz (the most commonly used condition in other studies) and a nearly zero drift ratio (ratio of lateral drift to forward distance), see the Supplementary Table S1 for a detailed discussion.

Additionally, it is worth mentioning that the velocity characteristics of our GH millirobots with various helix dimensions could be favorably predicted using a parametric dynamic model, as illustrated in Fig. 3c. A detailed description of this model is provided in Supplementary Note S1. Accuracy in model prediction ensures an efficient design-to-implementation process.

## Therapeutic drug delivery

To demonstrate the potential applications of our GH millirobots, we selected the therapeutic drug delivery application because precise and noncontact control of a millirobot's movement is crucial in liquid-containing environments with complex shapes. Figure 4 illustrates the therapeutic drug delivery process in the proposed application scenario. To validate the ability of the fabricated GH millirobots to conduct therapeutic drug delivery, we conducted in vivo manipulation of a GH millirobot in a rat's stomach. A GH millirobot, which was manipulated using a rotating magnetic field with a strength of 12 mT (far below the safety level with respect to human exposure[52,53]), was imaged in

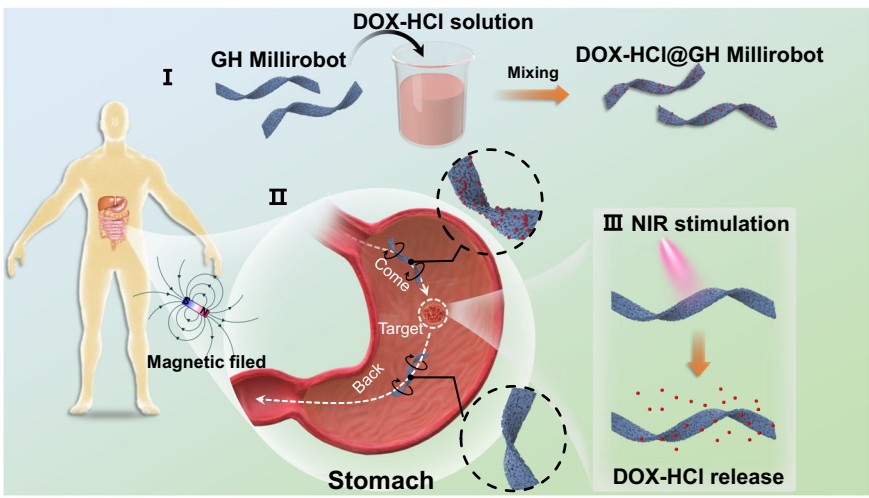

**Fig. 4 | Conceptual diagram of the use of GH millirobots for therapeutic drug delivery.** The combination of the GH millirobot with the DOX-HCl solution can result in the formation of a DOX-HCl@GH millirobot. The DOX-HCl@GH millirobot is capable of releasing DOX-HCl via near-infrared irradiation, which can be utilised for targeted therapy.

real-time by using a digital subtraction angiography (DSA) device. To ensure unambiguous identification and tracking, the path of the GH millirobot's movement had to be sufficiently long for the imaging system (which had resolution of ~200 μm); therefore, a rat was selected because of its adequately sized stomach. The results of the manipulation, which are shown in Fig. 5a and Supplementary Video 4, indicated that the manipulation successfully guided the GH millirobot to various targets in the rat's stomach. Moreover, the GH millirobot demonstrated its capability to execute targeted movements within the bladder filled with artificial urine in two scenarios (the details are shown in Supplementary S12 and Supplementary Video 5). Throughout these maneuvers, the microrobot effectively maintained a safe distance from the mucosal surface of the bladder wall, thus circumventing potential interference from surface structures—a common challenge encountered in the operation of wall-contact microrobots. Of significant importance, this avoidance of wall contact ensures the protection of non-cancerous cells from exposure to drugs during transport. These demonstrations underscore the advantages of GH millirobots in executing prolonged locomotion and facilitating drug transport within a physiological environment.

Subsequently, we examined the drug loading and release capabilities of the GH millirobots. Doxorubicin hydrochloride (DOX-HCl), which is a widely used anticancer drug with an aromatic molecular structure[54], was selected as the test drug because of its ability to readily bind to the GH millirobots through π–π, electrostatic, and hydrophobic interactions[55]. The loading process involved immersing the GH millirobots in DOX-HCl solution for a certain duration, after which the millirobots were rinsed with deionized water (see the Methods section for details; Fig. 4I). Confocal laser scanning microscopy imaging revealed bright red fluorescent spots on the millirobots (Fig. 5b), which confirmed that DOX-HCl molecules had successfully bound to various sections of the devices.

To trigger the drug release process, we employed near-infrared (NIR) irradiation as the stimulus because it does not involve contact, offers sufficient penetration depth, and causes little damage to organs[56]. As illustrated in Fig. 5d, DOX-HCl was rapidly released within 10 min upon NIR irradiation. We quantitatively evaluated the quantity of drug released by performing spectrophotometry (calibration curve is shown in Fig. 5c). The results suggested that approximately 0.03 μg of DOX-HCl was released from each DOX-HCl@GH millirobot during 5 min of NIR irradiation. Prolonged NIR irradiation beyond 10 min resulted in a rapid decrease in drug release, likely due to DOX-HCl

destabilization caused by high temperature associated with extended irradiation times. Furthermore, the quantity of drug released by each GH millirobot was approximately the same between batches, as indicated in Supplementary Note S2[57,58].

Finally, we evaluated the safety and efficacy of the GH millirobots' drug delivery by using a mouse model. The rat model was not applicable because of the lack of rats' immunodeficiency for tumor model construction. BALB/c nude mice, a widely adopted immunodeficient animal model, were selected, and gastric cancer was selected as the disease model because the method used to construct a tumor xenograft model in mice is technically mature. We first conducted safety examinations on healthy mice and found that the mice maintained a stable body weight even when they received numerous GH millirobots through gavage. The weights of their internal organs were almost identical to those of a control group (without GH millirobot loading, Supplementary Fig. S13; evidence from other biochemical analyses is displayed in Supplementary Figs. S14 and S15); this result suggested that the fabricated GH millirobots might function as safe drug carriers. Subsequently, the following three groups of mice with gastric cancer were treated: Group i, which was treated using pristine GH millirobots (i.e., not loaded with a drug not exposed to NIR irradiation); Group ii, which was treated using DOX-HCl-loaded GH millirobots but not exposed to NIR irradiation; and Group iii, which was treated using DOX-HCl@GH millirobots and exposed to NIR irradiation. The treatments lasted for 14 days. As shown in Supplementary Fig. S16b, the body weights of the mice in the three groups remained similar (with less than 10% deviation) at various stages of the treatment (the general decrease in the mice's body weights was caused by tumors). Furthermore, routine hematological analysis revealed that all hematological indices of the mice were normal after the treatments (Supplementary Fig. S17). These results collectively demonstrate the therapeutic safety of the treatment. In the meanwhile, the efficacy of the treatment was validated by the substantially smaller tumor weight in Group (iii) mice, as compared to Groups (i) and (ii), as illustrated in Fig. 5e and Supplementary Fig. S16a. Thus, the DOX-HCl-loaded GH millirobots exposed to NIR irradiation had an effective inhibitory effect on the mice's tumors. Biopsied tumor samples were subjected to immunohistochemical (IHC) staining (with Caspase-3, which appears brown) and general hematoxylin–eosin (H&E) staining to verify the treatment's efficiency. As shown in Fig. 5f, compared with the samples from the other groups, the Group iii samples exhibited more pronounced and larger brown areas in

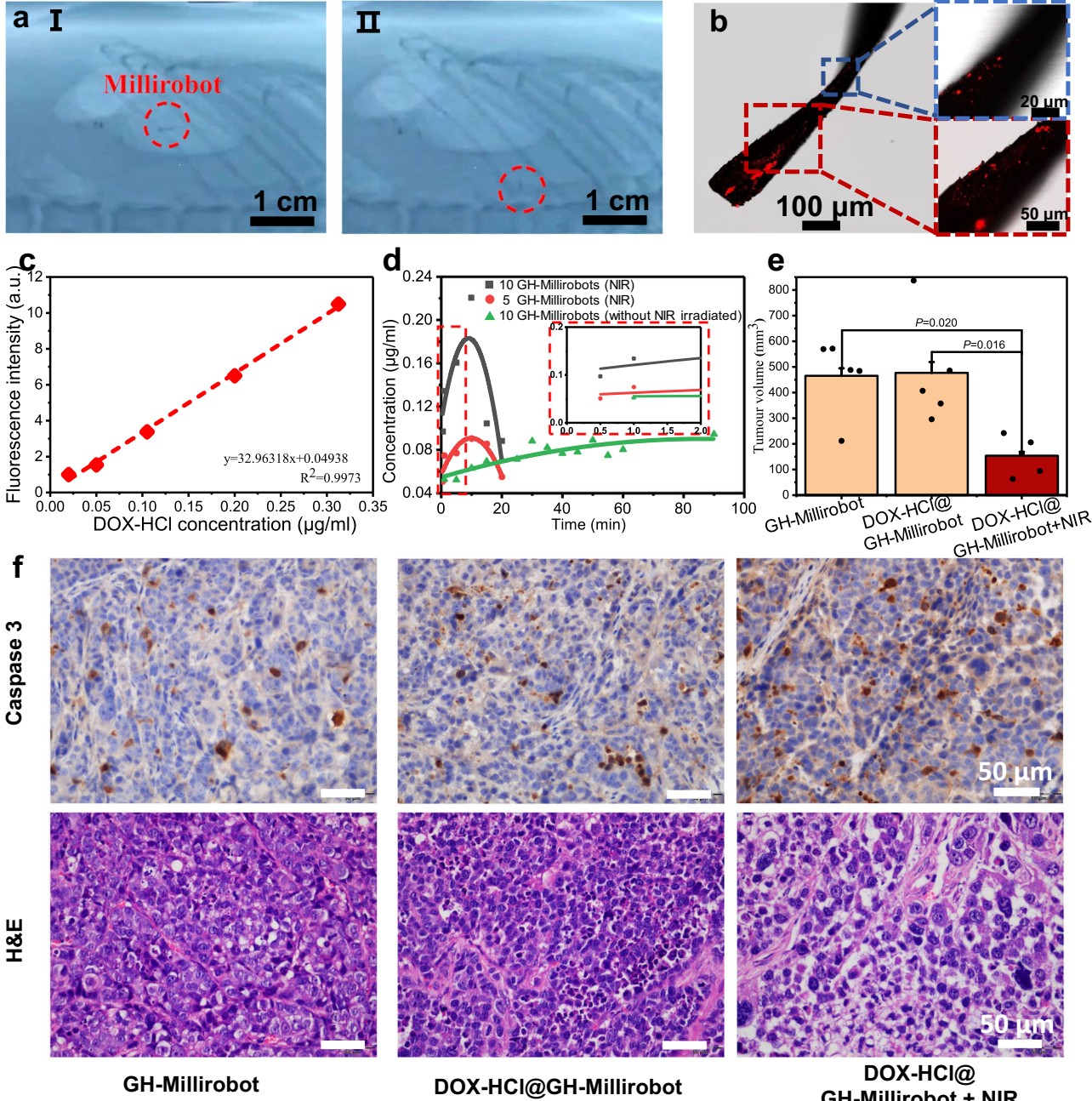

**Fig. 5 | Drug delivery and release by GH millirobots. a** Movement of a GH milli-robot in a rat stomach, where the millirobot was controlled by a rotating magnetic field with a strength of 12 mT. **b** Fluorescent image obtained under a confocal laser scanning microscope of a GH millirobot loaded with DOX-HCl. The experiment was repeated for three different batches and yield similar results. **c** Fluorescence intensity of DOX-HCl solutions of various concentrations (0.02, 0.05, 0.1, 0.2, and 0.3 μg/mL); the fluorescence intensity was measured using a fluorescence spec-trophotometer, and the concentrations of the released drug were fit through linear regression. a.u., arbitrary units. **d** Concentration of DOX-HCl released under NIR irradiation in 2 mL of deionized water. The curves were fitted using a quadratic Gaussian equation. The fluorescence counting was started 30 s after the initiation of DOX release at time point 0. Measurements were then taken at intervals of 1 min, 5 min, and subsequently every 5 min thereafter. The inset provides a magnified view of the initial 2-minute period. **e** Gastric tumor volume in three mice groups. The "DOX-HCl@GH-Millirobot" stand for GH-Millirobot loaded with DOX-HCl, ($n = 5$/group). Data are presented as the means ± s.e.m. **f** Gastric tumor biopsy images obtained under IHC staining with phosphorylated caspase (brown) and under H&E staining for the three mice groups. Scale bar: 50 μm.

IHC images and larger blank areas in H&E images, indicating more extensive tumor apoptosis and demonstrating that the treat-ment was effective. In conclusion, the aforementioned comprehensive results indicated that the GH millirobots carrying DOX-HCl that was to be released upon NIR irradiation were effective and safe for tumor therapy in mice. It's notable that the micro robots demonstrate func-tionality within the gastric cancer model due to the temporary fluidic environment that the stomach can create following water ingestion.

Nonetheless, these micro robots may find enhanced suitability for treating diseases within organs like the bladder, characterized by constant fluidic environments.

## Discussion
In this study, we demonstrated the high-throughput fabrication of porous GH millirobots by conducting linear scanning with triangular laser spots to convert PI into graphene. This processing method not

only induces the spontaneous twisting and peeling off of LIG sheets from a PI substrate but also provides the sheets a highly porous structure. As such, millirobot scaffolds with low density and high surface hydrophobicity are created, and the resultant millirobots remain fully suspended when being propelled magnetically. The nickel-coated GH millirobots exhibit impressive locomotion speed and excellent ability to track programmed trajectories, showing nearly zero deviation. By systematically varying the laser processing conditions, we demonstrated that the adopted fabrication method offers high throughput (77 millirobot scaffolds per second) and is highly controllable and versatile; various geometrical parameters linked to the locomotion properties of millirobots can be tuned using a parametric model. In particular, the combination of high production speed and low raw material cost makes it feasible to maintain the cost per millirobot device below one cent: US$0.01 (Supplementary Note S3). We validated the advantages of GH millirobots in the context of long-distance locomotion and drug delivery within a physiological environment by using gastric cancer drug delivery as an example. This study indicates the potential of GH millirobot technology to meet numerous application requirements simultaneously, such as performance, versatility, scalability, and cost-effectiveness requirements.

## Methods

### Ethics statement

This research complies with all relevant ethical regulations. This experiment performs in accordance with the requirements of the international accreditation code of the Association for Assessment and Accreditation of Laboratory Animal Care International and the policies of the Institutional Animal Care and Use Committee (No. B202309-5).

### Fabrication and characterization of GH millirobots

The PI film ($\approx 120\,\mu m$ thick) used to process the GH millirobots in the experiment was purchased from DuPont, USA. A piece of PI film was affixed to a glass substrate and then fixed to a laser system by using a self-developed bracket. The laser beam directly irradiated the PI film; it did not need to pass through the glass first.

The UV picosecond laser used in this study, including the optics and motion system, was developed in cooperation with Han's Laser. This laser could produce a light beam with a wavelength of 355 nm, a pulse duration of 10 ps, and a focused spot diameter of approximately 26.5 µm. The laser had a two-dimensional linear motion system (X- and Y-axis directions) and a Z-axis motion system. Along the X- and Y-axis, processing could be performed over a range of 30 mm × 30 mm; the maximum movement speed was 0.5 m/s. The distance between the focal plane of the laser beam and the working plane (i.e., the defocus distance) was adjusted using the Z-axis motion system. By controlling the aforementioned three laser parameters, we adjusted the energy density of the laser to within the window that would enable the porous helical LIG sheets to be removed from the PI film ($\sim 39{-}78\,J/cm^2$). All porous helical LIG sheets were formed at ambient temperature and pressure without the use of protective gases. A magnetron sputtering machine (Desk V TSV, Denton Vacuum) was employed for the deposition of nickel on the LIG sheets; this process was performed in an argon environment and by using a current of 90 mA for 16 min.

A scanning electron microscope (SU8220, Hitachi) was used to observe the surface morphology and microstructure of the LIG sheets. Moreover, Raman spectroscopy of the LIG sheets was conducted using a Raman microscope (LabRAM HR Evolution, HORIBA Jobin Yvon) with 532-nm laser excitation at room temperature. An optical microscope (CX41, OLYMPUS) was employed to examine the dimensions of the LIG sheets. The surface morphology and EDS spectra of the millirobots were observed after nickel plating by using a scanning electron microscope equipped with an energy-dispersive X-ray spectrometer (TESCAN MIRA LMS, the Czech Republic).

### Magnetic actuation and microscopy observation

A three-dimensional Helmholtz coil (PS-3HM400, Hunan Paisheng Technology Co., Ltd.) was used to generate a rotating magnetic field to actuate the GH millirobots. The frequency of the rotating magnetic field can vary between 1 Hz and 10 kHz. The microrobot was put into the liquid environment with different viscosities. And the solution inside the isolated pig bladder was artificial urine (purchased from local market) when the GH-Microrobot was moving in. The isolated porcine bladder was purchased from the local market. A high-speed camera (VW-600C, Keyence) was used to capture the movement of the millirobots. The forward velocity of a GH millirobot was determined from the time that its head left the scale line.

### Loading and release of DOX-HCl

Fifty GH millirobots (~2-mm long) were dispersed in 2 mL of DOX-HCl solution (concentration of 0.5 mg/mL); after 1 h, they were removed from the solution and rinsed several times with deionized water to remove the free DOX-HCl. Fluorescence images of a DOX-HCl@GH millirobot were captured using a confocal laser scanning microscope (Carl Zeiss, Germany). The release of DOX-HCl was stimulated by NIR irradiation. The NIR light was produced using an infrared lamp (BR125, PHILIPS); at 30 cm directly below the lamp, the average power density was $0.54\,W/cm^2$. The drug release experiments were conducted in 2 mL of deionized water. Three groups of millirobots were used; the first and second groups consisted of 10 and 5 millirobots, respectively, which were exposed to NIR irradiation; the third group consisted of 10 millirobots that were not exposed to NIR irradiation during their movement in water.

The fluorescence intensity of DOX-HCl solutions was measured using a fluorescence spectrophotometer (HORIBA Instruments Inc., USA); the excitation and emission wavelengths were 470 and 580 nm, respectively. The fluorescence intensity was normalized to obtain a calibration curve, on the basis of which the quantity of released DOX-HCl could be calculated after linear regression was performed. The concentrations of drug released by the millirobots under different NIR irradiation durations and in different groups were also calculated from the linear regression curves.

### In vivo manipulation and imaging of GH millirobots

The GH millirobots were mixed in 200 µL of water and delivered to the stomach of a rat through a gavage needle. The three-dimensional Helmholtz coil was used to manipulate the motion of the GH millirobot in the rat stomach. DSA (CGO-2100 Plus, Beijing Wandong Medical Technology Co., China) was performed to monitor and record the motion of the GH millirobots. The Institutional Animal Care and Use Committee (No: HT-FORM-IAC-001-1A, Huateng BioScience Co., Ltd.) approved all animal experimentation protocols of this study.

### In vivo safety test

Sixteen mice were equally divided into four groups, namely (i) a control group receiving no GH millirobot loading and groups loaded with (ii) 1000, (iii) 2000, and (iv) 4000 GH millirobots, loaded in each mouse through gavage on a daily basis for 7 consecutive days (the millirobots were naturally eliminated from the digestive system with food daily through excretion). The GH millirobots had a length of 2 mm. The mice were continuously observed for 14 days, and the hair, skin, eyes, respiration, behavioral activities, defecation, body weight, and food consumption of the mice were recorded each day. At the end of the 14th day, whole blood was collected for routine hematological analysis, and serum samples were collected for biochemical analysis.

### In vivo antitumor therapy

SGC-7901 gastric cancer cells were obtained from iCell Bioscience Inc. (Shanghai, China, Item No. icell-h277) and maintained in minimum essential medium supplemented with 10% fetal bovine serum (Gibco) at 37 °C and 5% $CO_2$. Subsequently, $4 \times 10^6$ SGC-7901 cells were

suspended in 200 µL of phosphate-buffered saline and injected subcutaneously into a 5-week-old female BALB/c nude mouse. This procedure was performed for 32 mice. When a subcutaneous tumor had grown to a diameter of approximately 10 mm, the tumor was collected and cut into small pieces with a diameter of 2–3 mm in precooled saline. The 32 BALB/c nude mice were anesthetized through intraperitoneal injection of 800–1200 mg/kg of 20% ursodiol. Subsequently, the abdominal cavity was opened, the stomach was gently pulled out, and the plasma membrane surface of the stomach on the anterior wall of the side with greater curvature was carefully scratched with ophthalmic scissors. A precut tumor block was then placed on the scratched area and fixed with OB adhesive, after which the stomach was returned to its original position, the abdomen was closed, and the stomach was sutured layer by layer. After this surgery, the animals were placed on a heating blanket until they had awoken, at which point they were returned to their cage. Penicillin (80,000 units) was injected daily for 3 days to prevent infection, and the mice were observed daily. After the tumors had grown for approximately 2 weeks, the animals were randomly divided into three groups, with eight mice per group: Group i, which was administered GH millirobots; Group ii, which was administered DOX-HCl@GH millirobots; and Group iii, which was administered DOX-HCl@GH millirobots induced by NIR irradiation. The drug therapy for each group was as follows: Group i, gavage with 2000 GH millirobots per mouse (200 µL suspension); Group ii, gavage with 2000 DOX-HCl@GH millirobots per mouse (200 µL suspension); and Group iii, gavage with 2000 DOX-HCl@GH millirobots per mouse (200 µL suspension) followed by NIR irradiation. In this irradiation, a light bulb was 30 cm from the mouse, the irradiation time was 5 min, and the dosing cycle was once a day. After 2 weeks, the mice were euthanized, and the tumors were harvested for IHC and H&E straining. At the end of the 14th day, whole blood was collected for routine hematological analysis. The temperature and humidity of the feeding environment meet the standard of ordinary animal house, and keep the temperature and humidity at 20 ~ 26 °C, 40 ~ 70%. The alternating time of day and night 12 h/12 h.

### Reporting summary

Further information on research design is available in the Nature Portfolio Reporting Summary linked to this article.

## Data availability

The data generated in this study are provided in the Supplementary Information/Source data file. Source data are provided with this paper.

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

## Acknowledgements

This work was supported by the National Key R&D Program of China (grant No. 2022YFB4701000, Y.C.), the National Natural Science Foundation of China (grant No. 51975127, Y.C., U20A6004, X.C.), and the Guangdong Basic and Applied Basic Research Foundation (grant No. 2022B1515120011, Y.C.). Guangzhou Basic and Applied Basic Research Foundation (grant No. 2024A04J6362, Y.C.).

## Author contributions

Y.C., Y.G., and B.X. designed the millirobots and conducted the experiments. Y.G., B.X., L.M., M.H., and H.L. developed the millirobot model. Y.C., Y.G., B.X., J.G., H.Z., L.M., Y.L., C.P.W, and N.Z. contributed to the scientific discussion and experimental design. F.J. and Y.J.L. conducted the cell culture and cytotoxicity experiments. Y.G. and B.X. produced 3D graphic illustrations and prepared the videos. Y.C., N.Z., and X.C. conceived the project. Finally, Y.C., Y.G., B.X., and N.Z. wrote the manuscript with contributions from all authors.

## Competing interests

The authors declare no competing interests.
