## [Peer Review File · Nature Communications]

Lightweight and drift-free magnetically actuated millirobots via asymmetric laser-induced grapheneREVIEWER COMMENTS

Reviewer #1 (Remarks to the Author):

The proposed method is about a novel way to produce microrobots with helical shape in high throughput while controlling and tuning multiple manufacturing parameters. Following comments can be considered to improve the paper further.

1. In the title, abstract, introduction and every other sections, authors claim that what they have produced and controlled is a "microrobot". If authors check the recent literature (last 15 years), most of the studies that use "microrobot" terminology have a robot that is smaller than 50 μm (entire size). If you look at your own reference number 6, that is a proper size to be called "microrobot". Your robots are actually in millimeter scale (1.2 mm). Even if it was 100s of micrometer, it would still be hard to call it microrobot. Therefore, author should consider different terminology for their robots such as small-scale, meso-scale or millimeter scale robots.
2. Can authors put explanation into their manuscript about what is the resolution of their manufacturing method in terms of lateral and depth printing capability (in 3D space). Can authors discuss what is the smallest helical robot that they could produce?
3. Most of the 49 references authors cite are not very new. Out of 49 references, only 6 of them are from last three years (2021-2023). This potentially demonstrates authors have not checked literature recently and therefore it is suggested that entire reference section should be updated with more recent examples from literature (last few years).

Reviewer #2 (Remarks to the Author):

Magnetically driven GH-Microrobots were designed and prepared by interesting ideas in this manuscript. They have precision targeting ability and can be fabricated in high-throughput and low-cost, which are promising in the field of drug delivery. However, I think the following points should be refined before publication.

Major comments:

1. Please mention the drug delivery function of the designed GH-Microrobots in the title of the manuscript.
2. What are the differences in properties of the two different chiral helices (left-handed and right-handed) presented in Fig. 2e?
3. The high-density microrobot and low-density microrobot were mentioned in Fig. 3a/d. It should be given more parameters of the high-density microrobot and low-density microrobot. Is there a threshold for low-density microrobots?
4. The average drug load of GH-Microrobot was calculated in the manuscript, but was it verified that the drug load of each GH-Microrobot was approximately the same between batches?
5. DOX-HCl@GH-Microrobots require an electrochemical workstation, a three-electrode system and KCl liquid environment to release DOX-HCl, how can authors realize the in-vivo release of drugs by electrochemical stimulation?
6. Is it possible to control the precise positioning of multiple DOX-HCl@GH-Microrobots simultaneously to release a sufficient amount of drug? The video should be provided
7. In cell experiments where electrical stimulation is given to release DOX-HCl, are three electrodes inserted into the cell solution? Is it necessary to add KCl? This part is not described in detail and should be improved.
8. In cytotoxicity assay, normal bladder cell assay should be added to further illustrate the effect of DOX-HCl@GH-Microrobots on normal cell activity.
9. When the robot moves in the urine environment, will the loaded drugs fall off spontaneously?
10. What is the healthy range of magnetic field strength? Why does the manuscript use the highest magnetic field strength compared to publications listed in Table S1?
11. Authors should explain the biological effects of helical LIG microsheets with different sizes. Are

there too long to perform tasks within the live body?

12. It is recommended that add animal experiments of the living body to demonstrate drug delivery and treatment efficiency.

Minor comments:

1. The word "Asymmetric" appears only in the title throughout the text, what is the author trying to reflect?

2. There was no 2.2 section, just 2.1, 2.3 and 2.4 sections.

Reviewer #3 (Remarks to the Author):

In this work authors prepare spiral graphene nanoribbons via laser-assisted process achieving such structures by laser profile forming. Authors further generate magnetic-actuated nanobots by applying thin nickel coating to those. These novel nanostructures are tested for motion in simulated bladder environments and are assessed for DOX delivery and release in bladder cancer cells. Although the work is critical for the field, novel and timely, it has several substantial deficiencies that would need to be addressed prior to publication.

- the language of the paper has to be corrected as there are many grammatical errors and the paper is currently hard to read.

- Figure 3 needs details about plot legend in Figure captions

- the description of DOX release experiments is missing and needs to be added. It is unclear if DOX is quenched on the nanoribbons, thus, it is not clear that the fluorescence analysis of its release are adequate. Full description of release assessment with fluorescence will be required to evaluate this.

- it is unclear if electrochemical trigger needed for release can damage cells. Thus, control experiments are needed with cells treated with nanobots and electrical trigger but without DOX.

Response to the comments on

Asymmetric laser-induced graphene enabling lightest and drift-free magnetically actuated millirobots

Reviewer #1 (Remarks to the Author):

The proposed method is about a novel way to produce microrobots with helical shape in high throughput while controlling and tuning multiple manufacturing parameters. Following comments can be considered to improve the paper further.

General response:

We thank the reviewer for the support to our work. Below please find our detailed answers to individual technical comments.

1. In the title, abstract, introduction and every other sections, authors claim that what they have produced and controlled is a "microrobot". If authors check the recent literature (last 15 years), most of the studies that use "microrobot" terminology have a robot that is smaller than 50 μm (entire size). If you look at your own reference number 6, that is a proper size to be called "microrobot". Your robots are actually in millimeter scale (1.2 mm). Even if it was 100s of micrometer, it would still be hard to call it microrobot. Therefore, author should consider different terminology for their robots such as small-scale, meso-scale or millimeter scale robots.

Response:

Thanks for the comment. We named the device “microrobot” based on the fact that its width and helical diameter are both in the micro-scale. We do appreciate the reviewer’s concern on the length of the robot reaching the millimeter scale. Therefore, we have replaced the name "microrobot" with "millirobot" in the revised manuscript, including the title.

2. Can authors put explanation into their manuscript about what is the resolution of their manufacturing method in terms of lateral and depth printing capability (in 3D space). Can authors discuss what is the smallest helical robot that they could produce?

Response:

The resolution of the laser manufacturing system is determined by the minimal laser spot size

(~26.5 μm) and the galvanometer's scanning accuracy (~10 μm). Note, however, that the formation of the millirobot is a spontaneous process involving both the formation and peeling-off of the LIG sheets. Therefore, the minimal size of the millirobot is determined not only by the resolution of the laser manufacturing system, but more importantly, by the intrinsic mechanical deformation property of the LIG sheets. In our experiment, the smallest millirobot size was about $86 \pm 4 \mu\text{m}$ in sheet width, $167 \pm 4 \mu\text{m}$ in helix diameter, and $986 \pm 6 \mu\text{m}$ in helix pitch (thus the minimal length of a millirobot containing at least one full helix pitch is around 1 mm).

Beyond the minimal sizes, the dimensions of the millirobots can be tuned by varying the laser spot size, scanning speed, and scanning length, with a tuning resolution of 13 μm , 47 μm , 18 μm , and 10 μm for the sheet width, helix diameter, pitch, and length, respectively (Supplementary **Fig S6**).

In the revised manuscript, we have provided the aforementioned details for the system processing resolution and minimal feature sizes of the millirobots, which are also shown below:

The smallest millirobot size was about $86 \pm 4 \mu\text{m}$ in sheet width, $167 \pm 4 \mu\text{m}$ in helix diameter, and $986 \pm 6 \mu\text{m}$ in helix pitch (thus the minimal length of a millirobot containing at least one full helix pitch is around 1 mm). Beyond these minimal sizes, the dimensions of the millirobots could be tuned by varying the laser spot size, scanning speed, and scanning length, with a tuning resolution of 13 μm , 47 μm , 18 μm , and 10 μm for the sheet width, helix diameter, pitch, and length, respectively (Supplementary **Figs. S5 & S6** and their corresponding discussions for details).

3. Most of the 49 references authors cite are not very new. Out of 49 references, only 6 of them are from last three years (2021-2023). This potentially demonstrates authors have not checked literature recently and therefore it is suggested that entire reference section should be updated with more recent examples from literature (last few years).

Response:

Thanks for the suggestion. We have updated the references by adding more studies published in recent years (2021-2023) in the Introduction section. The revisions are shown as follows:

In the past decade, proof-of-concept millirobots for drug/gene delivery^{4,8}, pollutant cleaning⁹⁻¹², and sensing¹³⁻¹⁵ applications have been demonstrated.

Many millirobots adopt a helical configuration¹⁷ as it can support both self-propulsion (such as chemical propulsion¹⁸) and external stimuli-based propulsion driven by a magnetic field^{19,20}, optical field²¹⁻²³ or acoustic field²⁴⁻²⁶. Several processing methods, including 3D direct laser lithography²⁷⁻²⁹, glancing angle deposition^{30,31}, biotemplate^{32,33}, laser ablation³⁴, and origami-based self-scrolling³⁵⁻³⁷, have been developed to create helical microstructures of photoresists, hydrogels, or metals for millirobot fabrication.

7. Chen, S. et al. Biodegradable microrobots for DNA vaccine delivery. *Adv. Healthc. Mater.* **12**, 2202921 (2023).
8. Lee, J. G. et al. Bubble-based microrobots with rapid circular motions for epithelial pinning and drug delivery. *Small.* **19**, 2300409 (2023).
11. Li, H. et al. Material-engineered bioartificial microorganisms enabling efficient scavenging of waterborne viruses. *Nat. Commun.* **14**, 4658 (2023).
12. Maria-Hormigos, R., Mayorga-Martinez, C. C. & Pumera, M. Soft magnetic microrobots for photoactive pollutant removal. *Small Methods.* **7**, 2201014 (2023).
14. Chen, H., Wang, Y., Liu, Y., Zou, Q. & Yu, J. Sensing of fluidic features using colloidal microswarms. *ACS Nano.* **16**, 16281-16291 (2022).
15. Wang, K. et al. Fluorescent self-propelled covalent organic framework as a microsensor for nitro explosive detection. *Appl. Mater. Today.* **19**, 100550 (2020).
21. Zhang, S. et al. Reconfigurable multi-component micromachines driven by optoelectronic tweezers. *Nat. Commun.* **12**, 5349 (2021).
22. Zhang, S. et al. Optoelectronic tweezers: a versatile toolbox for nano-/micro-manipulation. *Chem. Soc. Rev.* **51**, 9203-9242 (2022).
23. Wu, X. et al. Light-driven microdrones. *Nat. Nanotechnol.* **17**, 477 (2022).
26. Aghakhani, A. et al. High shear rate propulsion of acoustic microrobots in complex biological fluids. *Science advances.* **8**, m5126 (2022).
29. Lee, H. & Park, S. Magnetically actuated helical microrobot with magnetic nanoparticle retrieval and sequential dual-drug release abilities. *ACS Appl. Mater. Inter.* **15**, 27471-27485 (2023).
37. Nguyen, K. T. et al. A magnetically guided self-rolled microrobot for targeted drug delivery, real-time X-ray imaging, and microrobot retrieval. *Adv. Healthc. Mater.* **10**, 2001681 (2021).

Reviewer #2 (Remarks to the Author):

Magnetically driven GH-Microrobots were designed and prepared by interesting ideas in this manuscript. They have precision targeting ability and can be fabricated in high-throughput and low-cost, which are promising in the field of drug delivery. However, I think the following points should be refined before publication.

General response:

We thank the reviewer for the support to our work. Below please find our detailed answers to individual technical comments.

Major comments:

1. Please mention the drug delivery function of the designed GH-Microrobots in the title of the manuscript.

Response:

Thanks for the suggestion. Since the applications of the microrobots are not limited to drug delivery, and since the focus and main novelty of the manuscript is the fabrication method and new properties of the microrobots, rather than its application in drug delivery, we prefer not to put “drug delivery” in the title. However, we do appreciate the reviewer’s suggestion in highlighting this particular function of the microrobots. Therefore, we have added the description of the drug delivery function in both the abstract and conclusion of the manuscript. Note that in correspondence to Reviewer 1’s comment, the "GH-Microrobot" has now been replaced with "GH-Millirobot".

Abstract:

Importantly, such high-performance millirobots were fabricated at a speed of 77 scaffolds per second, demonstrating their potential in high-throughput and large-scale production. Using drug delivery for gastric cancer treatment as an example, we validated the advantages of the GH-Millirobots to conduct long-distance locomotion and drug transport in a physiological environment. This study demonstrates the potential of the GH-Millirobot technology in simultaneously fulfilling the application requirements on performances, versatility, scalability and cost-effectiveness.

Introduction

By systematically varying the laser processing conditions, we demonstrated that our fabrication method was high throughput (77 millirobot scaffolds per second) and versatile, providing tunability on various millirobot geometrical parameters, including the body length, sheet width, and the chirality, diameter, angle and pitch of the helical structures. Finally, we used therapeutic drug delivery for gastric cancer treatment as an example to demonstrate the advantages of the GH-Millirobots in conducting long-distance locomotion and drug transport in a physiological environment.

Discussion

In particular, the combination of high production speed and low raw-material cost enables the feasibility of maintaining the price per device at less than one cent (Supplementary Note S3). Furthermore, we have validated the advantages of GH-Millirobots in the context of long-distance locomotion and drug delivery within a physiological environment, using gastric cancer drug delivery as an example. This study demonstrates the potential of GH-Millirobot technology to simultaneously meet the application requirements in terms of performance, versatility, scalability and cost-effectiveness.

2. *What are the differences in properties of the two different chiral helices (left-handed and right-handed) presented in Fig. 2e?*

Response:

Helical LIG sheets with different chirality have the same material properties and microstructures and differ only in chirality. Accordingly, millirobots made from the helical LIG sheets of different chirality will have the same density but exhibit opposite rotating directions under the same magnetic field, resulting in the opposite moving directions, as shown in supplementary **Fig. S11** and **Video S3**. The explanation for this property difference is added to the revised manuscript as shown below:

In contrast, the low-density, fully suspended millirobot exhibited almost zero lateral drift during propulsion (**Fig. 3e**), demonstrating a remarkable ability in tracking the designated trajectories. Helical LIG sheets with different chirality had the same material properties and microstructures but differed only in chirality. Accordingly, millirobots made from the helical LIG sheets of different chirality had the same density but exhibited opposite rotating directions under the same magnetic field, resulting in opposite moving directions, as shown in supplementary **Fig. S11** and **Video S3**.

Fig. S11. Rotation of GH-Millirobots with right-handed (blue) and left-handed (red) chirality under the same magnetic field.

3. The high-density microrobot and low-density microrobot were mentioned in Fig. 3a/d. It should be given more parameters of the high-density microrobot and low-density microrobot. Is there a threshold for low-density microrobots?

Response:

Thanks for the comment. We classified the millirobots into the high-density and low-density groups, with the former showing a considerably higher density than that of water (e.g., 1 g/cm^3) and the latter showing a similar density as water. As such, the high-density millirobots tend to sink in water while the low-density ones can float in water. In **Section 2.2** we have provided the specific density values of the two millirobot types developed in this study, i.e., 1.023 g/cm^3 for the low-density millirobots and 1.42 g/cm^3 for the high-density millirobots.

Furthermore, in the revised manuscript we have added the detailed processing conditions for both types of millirobots, as shown below:

We first examined the impact of density on the trajectory accuracy of the GH-Millirobots. Here, by adjusting the laser processing parameters we fabricated two types of GH-Millirobots: a high-density (1.42 g/cm^3) millirobot that quickly sunk in the liquid environment and exhibited “on-ground” type translational motion (Fig. 3a, the processing details were shown in Supplementary Fig. S10), and a low-density (1.023 g/cm^3) millirobot that could be fully suspended in the liquid environment during propulsion (Fig. 3d).

4. The average drug load of GH-Microrobot was calculated in the manuscript, but was it verified that the drug load of each GH-Microrobot was approximately the same between batches?

Response:

Firstly, in each batch we performed the drug loading on the GH-Millirobots using the same experimental conditions including the drug concentration and solution volume. Secondly, since the efficacy of the therapy is determined by the amount of drug release, not the drug loading, for comparing the batch-to-batch performance variation of the millirobots, we characterized the drug release, instead of the loading, in each batch. In the drug release experiments, we randomly selected ten GH-Millirobots from Batch 1 and five GH-Millirobots from Batch 2, respectively, for comparison. Under the same NIR irradiation intensity, the ten GH-Millirobots from Batch 1 released $0.32 \mu\text{g}$ of drug in total and the five GH-Millirobots from batch 2 released $0.15 \mu\text{g}$ of drug in total, both to 2 mL of deionized water in 5 min, corresponding to an average release amount of $0.032 \mu\text{g}$ per GH-Millirobot for Batch 1 and $0.030 \mu\text{g}$ per GH-Millirobot for Batch 2. The results confirm that the drug release performance of the GH-Millirobots is consistent among different batches.

We have added the detailed discussion on the new drug release experiments in the revised manuscript and supplementary materials as shown below.

Note S2. Drug release of GM-Millirobot

To verify whether the drug release of each GH-Millirobot was consistent among different batches, we randomly selected ten GH-Millirobots from Batch 1 and five GH-Millirobots from Batch 2, respectively, for comparison (Fig. 4e). Under the same NIR irradiation intensity, the ten GH-Millirobots from Batch 1 released 0.32 μg of drug in total and the five GH-Millirobots from batch 2 released 0.15 μg of drug in total, both to 2 mL of deionized water in 5 min, corresponding to an average release amount of 0.032 μg per GH-Millirobot for Batch 1 and 0.030 μg per GH-Millirobot for Batch 2. The results confirm that the drug release performance of the GH-Millirobots is consistent among different batches.

Fig. 4. Drug delivery, transport and release of GH-Millirobot. e, Concentration of NIR induced DOX-HCl release to 2 mL deionized water. The curves were fitted using a quadratic Gaussian fit.

5. DOX-HCl@GH-Microrobots require an electrochemical workstation, a three-electrode system and KCl liquid environment to release DOX-HCl, how can authors realize the in-vivo release of drugs by electrochemical stimulation?

Response:

Thanks for the comment. Indeed, electrochemical stimulation is difficult to operate in practical applications. We used this method in the original manuscript as it is a convenient way to characterize the drug release performance, however, we agree with the reviewer on its limitation. In the revised manuscript, we have switched the stimulation method to near-infrared (NIR) irradiation, which has been proven as an effective tool of imaging and therapy of biological tissues and organs.

The experimental results related to the NIR irradiation-based simulation of DOX-HCl (drug) release are now shown in Fig. 4e, with the corresponding description provided in section 4.3, as well as follows:

To trigger the drug release process, we employed near-infrared (NIR) irradiation as the stimuli since it can be performed in a non-contact fashion, with sufficient penetration depth and little damage to the organs⁵⁹. As shown in **Fig. 4e**, the DOX-HCl was rapidly released within 10 min upon NIR irradiation. We further quantitatively evaluated the amount of the drug release by spectrophotometry (calibration curve shown in **Fig. 4d**), and the result suggested that there was approximately 0.03 μg of DOX-HCl released from each DOX-HCl@GH-Millirobot under NIR irradiation for 5 min. Prolonged NIR irradiation beyond 10 minutes resulted in a rapid decrease in drug release, likely due to DOX-HCl destabilization caused by high temperature associated with extended irradiation time. Furthermore, the drug release of each GH-Millirobot was approximately the same between batches as shown in supplementary **Note S2**.

4.3 Loading and release of DOX-HCl

Fifty GH-Millirobots (about 2 mm in length) were dispersed in 2 mL of DOX-HCl solution (0.5 mg/mL), removed after standing for 1 hour and rinsed several times with deionised water to remove the free DOX-HCl. Fluorescence images of a DOX-HCl@GH-Millirobot was taken by a confocal laser scanning microscope (Carl Zeiss, Germany). The release of DOX-HCl was stimulated by NIR irradiation. The NIR light was produced by an infrared lamp (BR125, PHILIPS) with an average power density of 0.54 W/cm² measured at 30 cm directly below the lamp. The drug release experiments were set up in 2 mL of deionized water and divided into three groups: group (i) and group (ii) consisted of ten millirobots and five millirobots, respectively, and were exposed to the NIR irradiation; group (iii) consists of ten millirobots but was not exposed to NIR irradiation during their movement in water.

The fluorescence intensity of DOX-HCl solutions was measured by a fluorescence spectrophotometer (HORIBA Instruments Incorporated, USA) with an excitation wavelength of 470 nm and emission of 580 nm. Then the fluorescence intensity was normalized to obtain a calibration curve, based on which the amount of released DOX-HCl can be calculated after linear regression. Drug concentrations released by the millirobots under different NIR irradiation duration and in different groups were also calculated from the linear regression curves.

Note S2. Drug release of GM-Millirobot

To verify whether the drug release of each GH-Millirobot was consistent among different batches, we randomly selected ten GH-Millirobots from Batch 1 and five GH-Millirobots from Batch 2, respectively, for comparison (**Fig. 4e**). Under the same NIR irradiation intensity, the ten GH-Millirobots from Batch 1 released 0.32 μg of drug in total and the five GH-Millirobots from batch 2 released 0.15 μg of drug in total, both to 2 mL of deionized water in 5 min, corresponding to an average release amount of 0.032 μg per GH-Millirobot for Batch 1 and 0.030 μg per GH-Millirobot for Batch 2. The results confirm that the drug release performance of the GH-Millirobots is consistent among different batches.

Fig. 4. Drug delivery, transport and release of GH-Millirobot e, Concentration of NIR induced DOX-HCl release to 2 mL deionized water. The curves were fitted using a quadratic Gaussian fit.

6. *Is it possible to control the precise positioning of multiple DOX-HCl@GH-Microrobots simultaneously to release a sufficient amount of drug? The video should be provided*

Response:

Thanks for the comment. Our current helical-shaped millirobots cannot be controlled as a group to precisely reach a target region. Controlling of multiple millirobots at different locations to reach a specific target can be achieved by changing the shape of the millirobots to spherical (our current on-going work), but this topic is beyond the scope of our current manuscript.

7. *In cell experiments where electrical stimulation is given to release DOX-HCl, are three electrodes inserted into the cell solution? Is it necessary to add KCl? This part is not described in detail and should be improved.*

Response:

The electrical stimulation release used in the original manuscript requires the insertion of three electrodes into the cell solution and the addition of 0.05 M KCl. As mentioned in the answers to Q4 and Q5, in the revised manuscript, we have changed to electrical stimulation to optical simulation.

8. *In cytotoxicity assay, normal bladder cell assay should be added to further illustrate the effect of DOX-HCl@GH-Microrobots on normal cell activity.*

Response:

Thanks for the suggestion. Since in the revised manuscript we have replaced all the cytotoxicity assay experiments with in-vivo experiments in living animals for validating the drug delivery performance, there are no more data plots/figures related to cytotoxicity assays. However, we understand the reviewer's general suggestion on providing control experiments on normal cells/animals, therefore, the following study was designed.

Firstly, we conducted a safety examination experiment using healthy mice, which showed that the mice maintained a stable body weight despite receiving a large number of GH-Millirobots (without DOX-HCl) via gavage and that the weight of their internal organs were almost the same as those of the control group without GH-Millirobot loading (**Fig. S12**; further evidences from other biochemical analysis are provided in **Fig. S13 & S14**), suggesting that the GH-Millirobots may work as a safe drug carrier.

Secondly, regarding the DOX-HCl effect. Note that DOX-HCl, as a strong cell-entering anti-tumor drug, is likewise harmful to normal cells/tissues. This is exactly why we need to precisely and quickly steer the drug-loaded millirobots to the target diseased tissues or cells, instead of letting them landing on normal cells. With this goal in mind, we treated three groups of mice with gastric cancer, including (i) one group treated with pristine GH-Millirobots (i.e., without drug loading), (ii) one group treated with DOX-HCl-loaded GH-Millirobots but without exposure to NIR irradiation, and (iii) one group treated with DOX-HCl@GH-Millirobot and exposed to NIR irradiation. The treatment lasted 14 days. As shown in **Fig. S15b**, the body weights of the mice in the three groups remained to be similar (i.e., with less than 10% deviation) at various stages of the treatment (the general decay of body weight is due to the presence of tumors in all mice). Furthermore, the routine hematological analysis shows that all the hematological indices of the mice appear to be normal after the treatment (**Fig. S16**). All these results demonstrate the therapeutic safety of the treatment. In the meanwhile, the efficacy of the treatment has also been validated by the significant smaller tumor volume and weight in the Group (iii) mice, as compared to Groups (i) and (ii), as shown in **Fig. 4f and Fig. S15a**.

We have added the related discussions in manuscript and supplementary materials as shown below. All the revisions in the manuscript are marked in **Yellow**.

As shown in Fig. S15b, the body weights of the mice in the three groups remained similar (i.e., with less than 10% deviation) at various stages of the treatment (the general decay of body weight is due to the presence of tumors in all mice). Furthermore, routine hematological analysis revealed that all hematological indices of the mice appeared normal after treatment (Fig. S16).

Fig. 4. Drug delivery, transport and release of GH-Millirobot. f, Gastric tumor volume in experimentally grouped mice. The “DOX-HCl@GH-Millirobot” stand for GH-Millirobot loaded with DOX-HCl. The error bars were standard error.

4.4.2 In vivo safety test

16 mice were equally divided into 4 groups, namely (i) control group with no GH-Millirobot loading, and groups with (ii) 1000, (iii) 2000, and (iv) 4000 GH-Millirobots, respectively, loaded in each mouse through gavage on a daily basis for 7 consecutive days (note that the millirobots were naturally eliminated from the digestive system with food daily). The GH-Millirobots were 2 mm in length. All the mice were continuously observed for 14 days, and the hair, skin, eyes, respiration, behavioral activities, defecation, body weight and food consumption of the mice were recorded each day. At the end of the 14th day whole blood was collected for routine hematological analysis and serum samples were collected for biochemical analysis.

4.4.3 In vivo antitumor therapy

SGC-7901 gastric cancer cells were obtained from iCell Bioscience Inc (Shanghai, China, Item No. icell-h277) and maintained in MEM medium supplemented with 10% fetal bovine serum (Gibco) at 37 °C and 5% CO₂. Then 4×10⁶ SGC-7901 cells were suspended in 200 μl PBS and injected subcutaneously into 5-week-old female BALB/c nude mice. When the subcutaneous tumor grown to a diameter of about 10 mm, the subcutaneous tumors were collected and cut into 2-3 mm diameter small pieces on pre-cooled saline. Thirty-two BALB/c nude mice were anesthetized by intraperitoneal injection of 20% ursodiol at a weight of 800-1200 mg/kg. Subsequently, the mice were anesthetized, and after opening the abdominal cavity, the stomach was gently pulled out, and the plasma membrane surface of the stomach on the anterior wall of the side of the greater curvature was carefully scratched with ophthalmic scissors, and the pre-cut tumor blocks were placed on the scratched area and fixed with OB adhesive, and then the stomachs were returned to the original position, the abdomen was closed, and the stomachs were sutured layer by layer. After surgery, the animals were placed on a heating blanket until the mice were awake and returned to the cage. 80,000 units of penicillin was injected daily for 3 days to prevent infection, and the mice were observed daily. After the tumors grew for about two weeks, the animals were randomly divided into three groups, eight mice for each group. Group A: GH-Millirobots, group B: DOX-HCl@GH-Millirobots, and group C: DOX-HCl@GH-Millirobots induced by NIR irradiation, respectively. Subsequently, drug therapy for each group: group A was subjected to gavage with 2000 GH-Millirobots per mice (200 μl suspension); group B was subjected to gavage with 2000 DOX-HCl@GH-Millirobots per mice (200 μl suspension); group C was subjected to gavage with 2000 DOX-HCl@GH-Millirobots per mice (200 μl suspension), followed by NIR irradiation, the light bulb was about 30 cm away from the mice, the irradiation time was 5 min, and the dosing cycle was once a day. After two weeks, the mice were euthanized, and the tumors were harvested for immunohistochemical and H&E staining analysis. At the end of the 14th day whole blood was collected for routine hematological analysis. This experiment was performed in accordance with the requirements of the international accreditation code of Association for Assessment and Accreditation of Laboratory Animal Care International (AAALAC) and the policies of the Institutional Animal Care and Use Committee (IACUC).

Fig. S12. Safety examination of GH-Millirobots on living mice. a, Body weight of mice during 7 consecutive day gavage of millirobots (control group, mice in other groups were in gavages of 1000, 2000 and 4000 GH-Millirobot, respectively) and follow-up 14 day monitoring. **b,** Comparison of organ weights of mice at the end of the safety experiment. All error bars were standard error.

Fig. S13. Safety examination of GH-Millirobots in living mice: routine hematological analysis and biopsies H&E staining of organs. a, Blood routine examination results of mice at 14th day. There were different numbers of pristine GH-Millirobots by gavage in mice and the safety test performed continuously for 7 days at a volume of 0.2 mL per mice in each day, and all the mice were continuously observed for 14 days. **b,** Tissue' biopsies (including heart, liver, spleen, lung, kidney) for H&E staining, which were collected from group iv (4000 GH-Millirobots) in safety experiments. All error bars were standard error.

9. When the robot moves in the urine environment, will the loaded drugs fall off spontaneously?

Response:

To characterize the fall-off of the loaded drugs, we monitored the drug concentration in the liquid environment where the DOX-HCl-loaded GH-Millirobots were suspended for 90 minutes. As shown by the green curve in **Fig. 4e**, only 0.188 μg DOX-HCl was dislodged from a total of ten GM-Millirobots after the 90-min movement. Note that the GH-Millirobots can move very fast with a maximum speed of 3.1 mm/s, therefore they can reach the target position in a very short time (usually less than thirty seconds). Therefore, there will be extremely little drug dislodged during the delivery process.

We have added the corresponding discussion in the supplementary materials as shown below.

Note S2. Drug release of GM-Millirobot

Besides, to characterize the fall-off of the loaded drugs, we monitored the drug concentration in the liquid environment where the DOX-HCl-loaded GH-Millirobots were suspended for 90 minutes. As shown by the green curve in **Fig. 4e**, only 0.188 μg DOX-HCl was dislodged from a total of ten GM-Millirobots after the 90-min movement. Note that the GH-Millirobots can move very fast with a maximum speed of 3.1 mm/s, thus they can reach the target position in a very short time (usually less than thirty seconds). Therefore, there will be extremely little drug dislodged during the delivery process.

Fig. 4. Drug delivery, transport and release of GH-Millirobot. e, Concentration of NIR induced DOX-HCl release to 2 mL deionized water. The curves were fitted using a quadratic Gaussian fit.

10. *What is the healthy range of magnetic field strength? Why does the manuscript use the highest magnetic field strength compared to publications listed in Table S1?*

Response:

For the safety range of the magnetic field strength, a previous study suggests that a magnetic field of up to 11.7 T could be safe to human⁵¹. Note that the magnetic resonance imaging systems in hospitals⁵² typically operate at a magnetic field strength of 1.5 T, 3 T, or 7 T, which are over 100 times higher than the 12 mT magnetic field used in the manuscript. Therefore, the 12 mT used in the manuscript is a safe magnetic field strength for humans.

As for why we used a higher magnetic field strength as compared to some other works, it is related to the low loading of the magnetic metal particles in the GH-Millirobots (i.e., the magnetic metal Ni layer taking only 6% volume fraction). The low loading ensures the full suspension mode of the of millirobots and thus enables fast movement and precise trajectory control; however, due to the reduced magnetic content, the magnetic field strength has to be increased.

We have added the discussions on the safety range and selection of magnetic field strength in the revised manuscript, as follows.

The GH-Millirobot, manipulated by a rotating magnetic field of 12 mT (well below the safety level with respect to human exposure^{55,56}), was imaged in real time with a digital subtraction angiography (DSA) device.

55. Elhanafy, A., Abuouf, Y., Elsagheer, S., Ookawara, S. & Ahmed, M. Effect of external magnetic field on realistic bifurcated right coronary artery hemodynamics. *Phys. Fluids*. **35**, 061903 (2023).
56. Shigemitsu, T. & Ueno, S. Biological and health effects of electromagnetic fields related to the operation of MRI/TMS. *SPIN*. **7**, 1740009 (2017).

11. Authors should explain the biological effects of helical LIG microsheets with different sizes. Are there too long to perform tasks within the live body?

Response:

Thanks for the suggestion. To address the concern on the biological effects, we carried out the following experiments. Firstly, we carried out in vivo safety experiments on mice to examine the biological effects of the helical LIG millisheets. 16 mice were equally divided into 4 groups, namely (i) control group with no GH-Millirobot loading, and groups with (ii) 1000, (iii) 2000, and (iv) 4000 GH-Millirobots, respectively, loaded in each mouse on a daily basis for 7 consecutive days (note that the millirobots were naturally eliminated from the digestive system with food daily). The GH-Millirobots were 2 mm in length. After the 7-day loading, the mice were further monitored for 14 days, and the hair, skin, eyes, respiration, behavioral activities, defecation, body weight and food consumption of the mice were recorded each day. The results show that all the mice exhibited good activity and nearly unchanged body weights till the end of the safety experiment and that the weight of their internal organs were almost the same as those of the control group without GH-Millirobot loading as shown in **Fig. S12**. Moreover, the results of the routine hematological and biochemical analysis on the mice also confirm that the GH-Millirobot loading does not have a significant impact on the mice conditions, as shown in **Fig. S13a and S14**, respectively.

Secondly, we verified whether the 2 mm long GH-Millirobots can be controlled to move to the target positions in organs such as stomach and bladder. In this experiment we delivered the GH-Millirobots to the rat stomach, manipulated it by a rotating magnetic field, and simultaneously imaged its movement by a digital subtraction angiography (DSA) device. **Fig. 4b** and **Video S4** demonstrated the GH-Millirobot was well follow the manipulation to various targets in the stomach.

The descriptions related to the two experiments are provided in **Sections 4.4.2** and **4.4.1**, respectively, and also shown as follows:

To validate the capability of the GH-Millirobots in performing such tasks, we conducted in vivo manipulation of a GH-Millirobot in a rat's stomach. The GH-Millirobot, manipulated by a rotating magnetic field of 12 mT (well below the safety level with respect to human exposure^{55,56}), was imaged in real time with a digital subtraction angiography (DSA) device. To ensure unambiguous identification and tracking, the moving path of the GH-Millirobots needed to be long enough for the imaging system (~200 μ m resolution), hence the choice of a rat due to its adequately sized stomach. The manipulation process shown in **Fig. 4b** and **Video S4** demonstrated that the GH-Millirobot could well follow the manipulation to various targets in the stomach.

Finally, we evaluated the safety and efficacy of the drug delivery treatment performed by the GH-Millirobots using mice as a model. Rat models were no longer applicable here due to their lack of immunodeficiency for tumor model construction. BALB/c nude mice, a widely adopted immunodeficient animal model, were chosen, and gastric cancer was selected as the disease model because its construction of tumor xenograft model in mice is technically mature. Firstly, we conducted the safety examinations using healthy mice, revealing that the mice maintained stable body weight despite receiving a large number of GH-Millirobots via gavage. The weights of their internal organs were almost identical to those of the control group without GH-Millirobot loading (**Fig. S12**; further evidences from other biochemical analysis are provided in **Fig. S13 & S14**), suggesting that GH-Millirobots may work as a safe drug carrier.

4.4.1 In vivo manipulation and imaging of GH-Millirobots

The GH-Millirobots were mixed in 200 μ L of water and delivered to the rat stomach via a gavage needle. The same three-dimensional Helmholtz coil was used for manipulating the motion of the GH-Millirobot in the rat stomach. Digital Subtraction Angiography (DSA) (CGO-2100 Plus, Beijing Wandong Medical Technology Co., China) was used to monitor and record the motion of the GH-Millirobot in the rat stomach. The Institutional Animal Care and Use Committee (IACUC) approved all animal experimentation protocols.

4.4.2 In vivo safety test

16 mice were equally divided into 4 groups, namely (i) control group with no GH-Millirobot loading, and groups with (ii) 1000, (iii) 2000, and (iv) 4000 GH-Millirobots, respectively, loaded in each mouse through gavage on a daily basis for 7 consecutive days (note that the millirobots were naturally eliminated from the digestive system with food daily). The GH-Millirobots were 2 mm in length. All the mice were continuously observed for 14 days, and the hair, skin, eyes, respiration, behavioral activities, defecation, body weight and food consumption of the mice were recorded each day. At the end of the 14th day whole blood was collected for routine hematological analysis and serum samples was

Fig. 4. Drug delivery, transport and release of GH-Millirobot. b, Movement of a GH-Millirobot in the rat's stomach controlled by a rotating magnetic field of 12 mT.

Fig. S12. Safety examination of GH-Millirobots on living mice. a, Body weight of mice during 7 consecutive day gavage of millirobots (control group, mice in other groups were in gavages of 1000, 2000 and 4000 GH-Millirobot, respectively) and follow-up 14 day monitoring. b, Comparison of organ weights of mice at the end of the safety experiment. All error bars were standard error.

Fig. S13. Safety examination of GH-Millirobots in living mice: routine hematological analysis and biopsies H&E staining of organs. a, Blood routine examination results of mice at 14th day. There were different numbers of pristine GH-Millirobots by gavage in mice and the safety test performed continuously for 7 days at a volume of 0.2 mL per mice in each day, and all the mice were continuously observed for 14 days. **b,** Tissues' biopsies (including heart, liver, spleen, lung, kidney) for H&E staining, which were collected from group iv (4000 GH-Millirobots) in safety experiments. All error bars were standard error.

12. It is recommended that add animal experiments of the living body to demonstrate drug delivery and treatment efficiency.

Response:

Thanks for the suggestion. As mentioned in our answers to Questions 8 and 11, we have performed several animal experiments on living bodies, as the reviewer suggested. To clearly

characterize the movement of the GH-millirobots in an organ of the living animal, a high-resolution real-time imaging system is needed. For our magnetically controlled millirobots, we selected digital subtraction angiography (DSA) for imaging as the magnetic field operation is not compatible with nuclear magnetic resonance imaging, while ultrasound imaging is limited by the probing depth and requirement of pressurized body contact. Since the resolution of DSA is approximately 200 μm , the moving path of the GH-Millirobots need to be long enough to allow the imaging system unambiguously identify and track the millirobots during their movement. Hence, we select rat for the experiments as its stomach is sufficiently large for arranging long movement for imaging. The manipulated process shown in **Fig. 4b** and **Video S4** demonstrate that the GH-Millirobot can well follow the manipulation to various targets in the stomach.

After confirming the controllability of the millirobots in animal organs through imaging, we then performed experiments to examine the safety and efficacy of the drug delivery treatment performed by the GH-Millirobots. For this part of the study, we used mice instead of rats because rats don't have immunodeficiency and therefore cannot be used for constructing tumor model for testing therapeutic drug delivery. BALB/c nude mouse, a widely adopted immunodeficient animal, was chosen as the animal model, and gastric cancer was selected as the disease model because its construction of tumor xenograft model in mice is technically mature. The detailed experimental findings are provided in the manuscript as well in the answers of Questions 8 & 11.

We have added the detailed experiment descriptions and corresponding discussions in the revised manuscript and supplementary materials as shown below.

To demonstrate the potential applications of the GH-Millirobots, we selected therapeutic drug delivery because it's criticality in precise and non-contact controlling of millirobot movement in complexly shaped liquid environments. **Fig. 4a** illustrates the therapeutic drug delivery process in the proposed application scenario. To validate the capability of the GH-Millirobots in performing such tasks, we conducted in vivo manipulation of a GH-Millirobot in a rat's stomach. The GH-Millirobot, manipulated by a rotating magnetic field of 12 mT (well below the safety level with respect to human exposure^{55,56}), was imaged in real time with a digital subtraction angiography (DSA) device. To ensure unambiguous identification and tracking, the moving path of the GH-Millirobots needed to be long enough for the imaging system (~200 μm resolution), hence the choice of a rat due to its adequately sized stomach. The manipulation process shown in **Fig. 4b** and **Video S4** demonstrated that the GH-Millirobot could well follow the manipulation to various targets in the stomach.

Finally, we evaluated the safety and efficacy of the drug delivery treatment performed by the GH-Millirobots using mice as a model. Rat models were no longer applicable here due to their lack of immunodeficiency for tumor model construction. BALB/c nude mice, a widely adopted immunodeficient animal model, were chosen, and gastric cancer was selected as the disease model because its construction of tumor xenograft model in mice is technically mature. Firstly, we conducted the safety examinations using healthy mice, revealing that the mice maintained stable body weight despite receiving a large number of GH-Millirobots via gavage. The weights of their internal organs were almost identical to those of the control group without GH-Millirobot loading (**Fig. S12**; further evidences from other biochemical analysis are provided in **Fig. S13 & S14**), suggesting that GH-Millirobots may work as a safe drug carrier. Subsequently, three groups of mice with gastric cancer were treated, including (i) one group treated with pristine GH-Millirobots (i.e., without drug loading), (ii) one group treated with DOX-HCl-loaded GH-Millirobots but without exposure to NIR irradiation, and (iii) one group treated with DOX-HCl@GH-Millirobot and exposed to NIR irradiation. The treatment lasted for 14 days. As shown in **Fig. S15b**, the body weights of the mice in the three groups remained similar (i.e., with less than 10% deviation) at various stages of the treatment (the general decay of body weight is due to the presence of tumors in all mice). Furthermore, routine hematological analysis revealed that all hematological indices of the mice appeared normal after treatment (**Fig. S16**). These results collectively demonstrated the therapeutic safety of the treatment. In the meanwhile, the efficacy of the treatment was validated by the significantly smaller tumor weight in Group (iii) mice, as compared to Groups (i) and (ii), as shown in **Fig. 4f and Fig. S15a**. This indicated that the GH-Millirobot loaded with DOX-HCl treatment induced by NIR had an effective inhibitory effect on tumors. Furthermore, the tumor biopsies were subjected to immunohistochemical (IHC) staining methods (Caspase-3, brown) and general hematoxylin-eosin (H&E) staining to verify the treatment efficiency. As shown in **Fig. 4g**, compared with other groups, Group iii displayed more pronounced and larger brown areas in IHC images and larger blank areas in HE images, indicating more tumor apoptosis and demonstrating effective treatment. In conclusion, these comprehensive results demonstrated that GH-Millirobots, carrying DOX-HCl to be released by NIR irradiation, was proved to be effective and safe for mouse tumor therapy.

4.4.3 In vivo antitumor therapy

SGC-7901 gastric cancer cells were obtained from iCell Bioscience Inc (Shanghai, China, Item No. icell-h277) and maintained in MEM medium supplemented with 10% fetal bovine serum (Gibco) at 37 °C and 5% CO₂. Then 4×10⁶ SGC-7901 cells were suspended in 200 μl PBS and injected subcutaneously into 5-week-old female BALB/c nude mice. When the subcutaneous tumor grown to a diameter of about 10 mm, the subcutaneous tumors were collected and cut into 2-3 mm diameter small pieces on pre-cooled saline. Thirty-two BALB/c nude mice were anesthetized by intraperitoneal injection of 20% ursodiol at a weight of 800-1200 mg/kg. Subsequently, the mice were anesthetized, and after opening the abdominal cavity, the stomach was gently pulled out, and the plasma membrane surface of the stomach on the anterior wall of the side of the greater curvature was carefully scratched with ophthalmic scissors, and the pre-cut tumor blocks were placed on the scratched area and fixed with OB adhesive, and then the stomachs were returned to the original position, the abdomen was closed, and the stomachs were sutured layer by layer. After surgery, the animals were placed on a heating blanket until the mice were awake and returned to the cage. 80,000 units of penicillin was injected daily for 3 days to prevent infection, and the mice were observed daily. After the tumors grew for about two weeks, the animals were randomly divided into three groups, eight mice for each group. Group A: GH-Millirobots, group B: DOX-HCl@GH-Millirobots, and group C: DOX-HCl@GH-Millirobots induced by NIR irradiation, respectively. Subsequently, drug therapy for each group: group A was subjected to gavage with 2000 GH-Millirobots per mice (200 μl suspension); group B was subjected to gavage with 2000 DOX-HCl@GH-Millirobots per mice (200 μl suspension); group C was subjected to gavage with 2000 DOX-HCl@GH-Millirobots per mice (200 μl suspension), followed by NIR irradiation, the light bulb was about 30 cm away from the mice, the irradiation time was 5 min, and the dosing cycle was once a day. After two weeks, the mice were euthanized, and the tumors were harvested for immunohistochemical and H&E staining analysis. At the end of the 14th day whole blood was collected for routine hematological analysis. This experiment was performed in accordance with the requirements of the international accreditation code of Association for Assessment and Accreditation of Laboratory Animal Care International (AAALAC) and the policies of the Institutional Animal Care and Use Committee (IACUC).

Minor comments:

1. The word "Asymmetric" appears only in the title throughout the text, what is the author trying to reflect?

Response:

Thanks for pointing out the issue. LIG is typically performed by a circular shaped laser beam and yields flat patterns. In this work, by changing the laser beam to an "asymmetric" shape, we obtained the helical LIG structures. In the revised manuscript, we have added "asymmetric" to a few places, where appropriate, to echo the title and highlight the unique property of the processing

laser beam. The modified parts are shown below:

Here, we developed a series of graphene-based helical millirobots (GH-Millirobots) by introducing **asymmetric** light pattern distortion in a laser-induced polymer-to-graphene conversion process, which spontaneously induced twisting and peeling-off of graphene sheets from a polymer substrate.

In our fabrication platform, however, we intentionally tilted one of the F- θ lenses, shifting the laser beam away from the original plane focusing surface, inducing **asymmetric** distortion of the laser spot on the working plane. By properly adjusting the distance between the working plane and the focal plane (i.e., the defocusing distance), a laser spot with a rounded triangular shape was obtained at the working plane, as illustrated in **Fig. 1a** and **Supplementary Fig. S1b**.

Since the gas production was induced by the laser, the geometry of the laser spot influenced the final configuration of the LIG sheet. For instance, if a triangular laser spot **was asymmetric along the laser scanning direction, such as** aligning the laser scanning direction along one of the edges of the triangle (e.g., edge III in **Fig. 1b**), significantly more gas was formed at the bottom edge of the LIG sheet due to more intense laser exposure compared to its top edge.

2. There was no 2.2 section, just 2.1, 2.3 and 2.4 sections.

Response:

Thanks for point out the mistake. We have corrected the section numbers as follows:

2.1 Processing and characterizations of porous helical LIG millisheets

2.2 Manipulation and characterization of graphene-based helical millirobots

2.3 Therapeutic drug delivery

Reviewer #3 (Remarks to the Author):

In this work authors prepare spiral graphene nanoribbons via laser-assisted process achieving such structures by laser profile forming. Authors further generate magnetic-actuated nanobots by applying thin nickel coating to those. These novel nanostructures are tested for motion in simulated bladder environments and are assessed for DOX delivery and release in bladder cancer cells. Although the work is critical for the field, novel and timely, it has several substantial deficiencies that would need to be addressed prior to publication.

General response:

We thank the reviewer for the support to our work. Below please find our detailed answers to individual technical comments.

1. the language of the paper has to be corrected as there are many grammatical errors and the paper is currently hard to read.

Response:

Thanks for your suggestion. We have carefully checked the manuscript, corrected grammatical mistakes and improved the English writing.

2. Figure 3 needs details about plot legend in Figure captions

Response:

Thanks for the comment. We have added the required details to the caption of Figure 3. Note that in correspondence to Reviewer 1's comment, the "GH-Microrobot" has now been replaced with "GH-Millirobot". The revisions are also shown below:

3. the description of DOX release experiments is missing and needs to be added. It is unclear if DOX is quenched on the nanoribbons, thus, it is not clear that the fluorescence analysis of its release are adequate. Full description of release assessment with fluorescence will be required to evaluate this.

Response:

Thanks for the suggestion. In the revised manuscript we have provided the full description of the drug release assessment of the millirobots. Note that in correspondence to Reviewer 2’s comment, the electrochemical stimulation of drug release has now been replaced with NIR light stimulation. The drug release assessment by NIR stimulating has been full described as follow.

First, we obtained the concentration-dependent fluorescence characteristics of DOX-HCl by measuring the fluorescence intensity of the solution samples with the DOX-HCl concentration of 0.02, 0.05, 0.1, 0.2, and 0.3 $\mu\text{g/mL}$, respectively. The relationship between the drug concentration

and fluorescence intensity was obtained by linear regression, as shown in **Fig. 4d**.

Subsequently, we verified the drug release of each GH-Millirobot was approximately the same between batches. In each batch we performed the drug loading on the GH-Millirobots using the same experimental conditions including the drug concentration and solution volume. Since the efficacy of the therapy is determined by the amount of drug release, not the drug loading, for comparing the batch-to-batch performance variation of the millirobots, we characterized the drug release in each batch. In the drug release experiments, we randomly selected ten GH-Millirobots from Batch 1 and five GH-Millirobots from Batch 2, respectively, for comparison. Under the same NIR irradiation intensity, the ten GH-Millirobots from Batch 1 released 0.32 μg of drug in total and the five GH-Millirobots from batch 2 released 0.15 μg of drug in total, both to 2 mL of deionized water in 5 min, corresponding to an average release amount of 0.032 μg per GH-Millirobot for Batch 1 and 0.030 μg per GH-Millirobot for Batch 2. The results confirm that the drug release performance of the GH-Millirobots is consistent among different batches.

Besides, to characterize the fall-off of the loaded drugs, we monitored the drug concentration in the liquid environment where the DOX-HCl-loaded GH-Millirobots were suspended for 90 minutes. As shown by the green curve in **Fig. 4e**, only 0.188 μg DOX-HCl was dislodged from a total of ten GM-Millirobots after the 90-min movement. Note that the GH-Millirobots can move very fast with a maximum speed of 3.1 mm/s, therefore they can reach the target position in a very short time (usually less than thirty seconds). Therefore, there will be extremely little drug dislodged during the delivery process.

The corresponding fluorescence release analysis and experimental procedures are shown in **section 4.3**. The revisions are also shown below:

To trigger the drug release process, we employed near-infrared (NIR) irradiation as the stimuli since it can be performed in a non-contact fashion, with sufficient penetration depth and little damage to the organs⁵⁹. As shown in **Fig. 4e**, the DOX-HCl was rapidly released within 10 min upon NIR irradiation. We further quantitatively evaluated the amount of the drug release by spectrophotometry (calibration curve shown in **Fig. 4d**), and the result suggested that there was approximately 0.03 μg of DOX-HCl released from each DOX-HCl@GH-Millirobot under NIR irradiation for 5 min. Prolonged NIR irradiation beyond 10 minutes resulted in a rapid decrease in drug release, likely due to DOX-HCl destabilization caused by high temperature associated with extended irradiation time. Furthermore, the drug release of each GH-Millirobot was approximately the same between batches as shown in supplementary **Note S2**.

4.3 Loading and release of DOX-HCl

Fifty GH-Millirobots (about 2 mm in length) were dispersed in 2 mL of DOX-HCl solution (0.5 mg/mL), removed after standing for 1 hour and rinsed several times with deionised water to remove the free DOX-HCl. Fluorescence images of a DOX-HCl@GH-Millirobot was taken by a confocal laser scanning microscope (Carl Zeiss, Germany). The release of DOX-HCl was stimulated by NIR irradiation. The NIR light was produced by an infrared lamp (BR125, PHILIPS) with an average power density of 0.54 W/cm² measured at 30 cm directly below the lamp. The drug release experiments were set up in 2 mL of deionized water and divided into three groups: group (i) and group (ii) consisted of ten millirobots and five millirobots, respectively, and were exposed to the NIR irradiation; group (iii) consists of ten millirobots but was not exposed to NIR irradiation during their movement in water.

The fluorescence intensity of DOX-HCl solutions was measured by a fluorescence spectrophotometer (HORIBA Instruments Incorporated, USA) with an excitation wavelength of 470 nm and emission of 580 nm. Then the fluorescence intensity was normalized to obtain a calibration curve, based on which the amount of released DOX-HCl can be calculated after linear regression. Drug concentrations released by the millirobots under different NIR irradiation duration and in different groups were also calculated from the linear regression curves.

Note S2. Drug release of GM-Millirobot

To verify whether the drug release of each GH-Millirobot was consistent among different batches, we randomly selected ten GH-Millirobots from Batch 1 and five GH-Millirobots from Batch 2, respectively, for comparison (Fig. 4e). Under the same NIR irradiation intensity, the ten GH-Millirobots from Batch 1 released 0.32 μg of drug in total and the five GH-Millirobots from batch 2 released 0.15 μg of drug in total, both to 2 mL of deionized water in 5 min, corresponding to an average release amount of 0.032 μg per GH-Millirobot for Batch 1 and 0.030 μg per GH-Millirobot for Batch 2. The results confirm that the drug release performance of the GH-Millirobots is consistent among different batches.

Besides, to characterize the fall-off of the loaded drugs, we monitored the drug concentration in the liquid environment where the DOX-HCl-loaded GH-Millirobots were suspended for 90 minutes. As shown by the green curve in Fig. 4e, only 0.188 μg DOX-HCl was dislodged from a total of ten GM-Millirobots after the 90-min movement. Note that the GH-Millirobots can move very fast with a maximum speed of 3.1 mm/s, thus they can reach the target position in a very short time (usually less than thirty seconds). Therefore, there will be extremely little drug dislodged during the delivery process.

4. it is unclear if electrochemical trigger needed for release can damage cells. Thus, control experiments are needed with cells treated with nanobots and electrical trigger but without DOX.

Response:

Thanks for the comment. Note that in correspondence to Reviewer 2's comment, in the revised manuscript, we have switched the stimulation method to near-infrared (NIR) irradiation, which has been proven as an effective tool of imaging and therapy of biological tissues and organs. Furthermore, the in-vitro cell experiments are replaced with animal experiments on living bodies. And the results demonstrated that GH-Millirobots carried with DOX-HCl released by NIR irradiation is an effective and safe treatment for mouse tumor therapy. The corresponding animal experiments analysis and experimental procedures are shown in **section 2.3**.

We have added the description of safety of NIR in the manuscript. The revisions are also shown below:

To trigger the drug release process, we employed near-infrared (NIR) irradiation as the stimuli since it can be performed in a non-contact fashion, with sufficient penetration depth and little damage to the organs⁵⁹. As shown in **Fig. 4e**, the DOX-HCl was rapidly released within 10 min upon NIR irradiation. We further quantitatively evaluated the amount of the drug release by spectrophotometry (calibration curve shown in **Fig. 4d**), and the result suggested that there was approximately 0.03 μg of DOX-HCl released from each DOX-HCl@GH-Millirobot under NIR irradiation for 5 min. Prolonged NIR irradiation beyond 10 minutes resulted in a rapid decrease in drug release, likely due to DOX-HCl destabilization caused by high temperature associated with extended irradiation time. Furthermore, the drug release of each GH-Millirobot was approximately the same between batches as shown in supplementary **Note S2**.

59. Yang, J. et al. Beyond the Visible: Bioinspired Infrared Adaptive Materials. *Adv. Mater.* **33**, 2004754 (2021).

REVIEWER COMMENTS

Reviewer #3 (Remarks to the Author):

Authors have answered some of the questions in their revision, however two minor points remain:
- Manuscript text still contains plenty of grammatical errors and extra long sentences in the introduction, which makes it hard to read and comprehend. Definitely not Nature Comm level writing.

- In the determination of release it is not clear how the authors account for DOX fluorescence at 0 time point. Since it is non-zero, there is some DOX fluorescence due to the drug sitting on the GH-Millirobots. This fluorescence is not due to released DOX and should be corrected for. Moreover, it will decrease as DOX would be released decreasing the baseline for each following measurement.

Reviewer #4 (Remarks to the Author):

For Reviewer 1 - comment 3:

Although new references were added, they do not accurately reflect the state of the art and as such, may result in unfair comparisons.

For instance, in lines 66-67, the authors claim "setting new records in terms of both speed and accuracy for the millirobots 16, 32, 42-44".

This is unfair, as only one of the reference (Ref 16) can be considered new (2019). The rest comes from the early 2010s. Subsequent works in the field have demonstrated significantly increases in speed and accuracy, much higher than the 2.64 blps reported. (Review article on state of the art for microswimmers: 10.1016/j.physrep.2018.10.007)

I would recommend that this sentence/ superlatives be removed to avoid misleading readers.

Other comments:

As it was stated that the density and surface wettability could also be precisely controlled with the proposed method (Lines 142-143), could the authors also provide a range and resolution of values? Only the range for the density of the robots are provided in the current manuscript.

For Reviewer 2 comments:

The paper is proposing a high-throughput method to produce helical microrobots. The intention of the authors was probably to only show readers the possible applications of the proposed method (i.e. drug delivery) in the demo. The manuscript, in its current form, emphasises too much on the demonstration and not on the novelty of the proposed method – almost a third of the paper is talking about the in vivo experiments.

In my opinion, the authors should have focused more on the novelty of the fabrication method – low density, high surface hydrophobicity and high throughput (in addition to the porosity which enables it to carry drugs). As it stands, the in vivo experiments conducted are too simplistic and in fact raises more questions than it answers. Further questions I have about the paper (specific to the demo) are:

1) The robots were not actually navigated to the target location. It seemed like the mice (with tumours) were force-fed a solution of microrobots. Locomotion demo (video 4), although present, was conducted in a rat. Although the authors did mention in the response letter that "Rat models were no longer applicable here due to their lack of immunodeficiency for tumor model construction. BALB/c nude mice, a widely adopted immunodeficient animal model, were chosen...", with the given resolution "When the defocus distance increased from 3 mm to 9 mm, the sheet width increased from $86 \pm 4 \mu\text{m}$ to the largest value of $117 \pm 3 \mu\text{m}$ at 8 mm and then slightly decreased to $110 \pm 6 \mu\text{m}$, the helix diameter correspondingly increased from $167 \pm 4 \mu\text{m}$ to $498 \pm 10 \mu\text{m}$.", it appears to me that it might be possible to fabricate a robot which can locomote to the tumour, while still being compatible with their imaging system (200 um resolution).

2) Moreover, if the robot is intended to be used for treating gastric tumours, then what is the

purpose of using a swimmer? I would imagine that microswimmers would be more suited for diseases in fluidic environments (i.e. full bladder, eye, blood vessels).

3) Is it possible to observe the tumour and locomote the robot accurately to the target?

4) Is the drug carried by a single helical swimmer sufficient? And if it is not, then are there strategies to overcome the insufficient drug dose, considering that the authors already ruled out swarm control?

Having said that, I think that the authors did satisfactorily address Reviewer 2's comments.

Response to the comments on

Asymmetric laser-induced graphene enabling lightest and drift-free magnetically actuated millirobots

Reviewer #3 (Remarks to the Author):

Authors have answered some of the questions in their revision, however two minor points remain:

General response:

We thank the reviewer for the support to our work. Below please find our detailed answers to the individual technical comments.

- Manuscript text still contains plenty of grammatical errors and extra long sentences in the introduction, which makes it hard to read and comprehend. Definitely not Nature Comm level writing.

Response:

Thank you for your feedback. We have taken your suggestions into consideration and conducted a thorough revision of the manuscript to address the grammatical errors. Additionally, we have engaged a professional English editing service to enhance the overall quality of the English writing in the manuscript.

- In the determination of release it is not clear how the authors account for DOX fluorescence at 0 time point. Since it is non-zero, there is some DOX fluorescence due to the drug sitting on the GH-Millirobots. This fluorescence is not due to released DOX and should be corrected for. Moreover, it will decrease as DOX would be released decreasing the baseline for each following measurement.

Response:

Thank you for your feedback, and we apologize for any confusion regarding the method used to determine drug release in our original manuscript. To address this, we want to clarify that the initiation of DOX fluorescence counting in Figure 4e does not coincide with the 0-time point, but rather begins 30 seconds after the 0 second mark. To elucidate this further, we have included an

enlarged plot focusing on the initial 2-minute timeframe as an inset within Figure 4e. Basically, after 30 seconds the GH millirobots had released sufficient DOX to yield a discernible fluorescence signal. We have added more detailed description on the experimental process in the caption of Figure 4e for clarity.

Reviewer #4 (Remarks to the Author):

For Reviewer 1 - comment 3:

Although new references were added, they do not accurately reflect the state of the art and as such, may result in unfair comparisons.

For instance, in lines 66-67, the authors claim “setting new records in terms of both speed and accuracy for the millirobots 16, 32, 42-44”.

This is unfair, as only one of the reference (Ref 16) can be considered new (2019). The rest comes from the early 2010s. Subsequent works in the field have demonstrated significantly increases in speed and accuracy, much higher than the 2.64 blps reported. (Review article on state of the art for microswimmers: 10.1016/j.physrep.2018.10.007)

I would recommend that this sentence/ superlatives be removed to avoid misleading readers.

Response:

Many thanks you for your feedback. We appreciate your concerns regarding the relevance of the references cited in our manuscript. Our comparison was specifically focused on the speed of helical microswimmers under a magnetic-field frequency of 10Hz, as indicated in our discussion. While we acknowledge that subsequent works in the field have demonstrated advancements in speed

and accuracy, it's important to note that these achievements typically involve different microswimmer structures, driving methods, or operating frequencies, which are not directly comparable to our study.

In response to your suggestions, we have updated our references to include more recent works (10.1021/acsami.3c01087; 10.3390/pharmaceutics14112393) to provide a more comprehensive overview of the current state of the art. Furthermore, we have followed the suggestion of the reviewer to delete the statement of “setting new records in terms of both speed and accuracy for the millirobots” and refined the caption of Figure 3f: “f, Comparison of the swimming velocity and drift ratio of magnetically actuated helical millirobots”. We hope these clarifications address your concerns and provide a clearer perspective on our findings.

Other comments:

As it was stated that the density and surface wettability could also be precisely controlled with the proposed method (Lines 142-143), could the authors also provide a range and resolution of values? Only the range for the density of the robots are provided in the current manuscript.

Response:

Thank you for your suggestion. Given the micron size and helical structure of the microrobots, direct measurement of their surface wettability is very challenging. Instead, we employed planar laser-induced graphene (LIG) samples processed under the same laser power conditions to assess surface wettability variations. Note that the switch from helical to planar LIG can be simply accomplished by changing the laser spot shape from triangular to round. Our experimental findings (shown in Figure R1) demonstrate the ability to modulate contact angles within the range of 94°-131° by adjusting laser power levels. We have added Figure R1 as Figure S2c in the Supporting

Information.

For Reviewer 2 comments:

The paper is proposing a high-throughput method to produce helical microrobots. The intention of the authors was probably to only show readers the possible applications of the proposed method (i.e. drug delivery) in the demo. The manuscript, in its current form, emphasises too much on the demonstration and not on the novelty of the proposed method – almost a third of the paper is talking about the in vivo experiments. In my opinion, the authors should have focused more on the novelty of the fabrication method – low density, high surface hydrophobicity and high throughput (in addition to the porosity which enables it to carry drugs). As it stands, the in vivo experiments conducted are too simplistic and in fact raises more questions than it answers. Further questions I have about the paper (specific to the demo) are:

General response:

We appreciate the reviewer's insightful comments. Indeed, our primary focus lies in presenting the innovative fabrication method for helical microrobots, as originally intended. Following initial peer review, we were prompted by one of the reviewers to conduct in vivo experiments on animal models to validate the microrobots' performance within living organisms. Consequently, the current version of the manuscript includes expanded discussions on the potential applications of the microrobots.

Understanding the reviewer's concerns, we recognize the need to strike a balance between highlighting the novelty of the fabrication method and providing adequate insights into the in vivo experiments. We acknowledge the reviewer's efforts in pinpointing areas for improvement and are grateful for the detailed questions provided, which have aided us in refining the descriptions pertaining to the in vivo experiments. Please find below our detailed responses to each technical comment provided.

1) The robots were not actually navigated to the target location. It seemed like the mice (with tumours) were force-fed a solution of microrobots. Locomotion demo (video 4), although present, was conducted in a rat. Although the authors did mention in the response letter that “Rat models were no longer applicable here due to their lack of immunodeficiency for tumor model construction. BALB/c nude mice, a widely adopted immunodeficient animal model, were chosen... “, with the given resolution “When the defocus distance increased from 3 mm to 9 mm, the sheet width increased from $86 \pm 4 \mu\text{m}$ to the largest value of $117 \pm 3 \mu\text{m}$ at 8 mm and then slightly decreased to $110 \pm 6 \mu\text{m}$, the helix diameter correspondingly increased from $167 \pm 4 \mu\text{m}$ to $498 \pm 10 \mu\text{m}$.”, it appears to me that it might be possible to fabricate a robot which can locomote to the tumour, while still being compatible with their imaging system (200 um resolution).

Response:

Thanks for the comment. Indeed, our in vivo experiments involved administering a solution of microrobots to mice with tumors rather than directly navigating the microrobots to the target location. However, we did present a locomotion demonstration in a rat in Video 4.

Furthermore, in our original submission, we also conducted experiments involving ex vivo manipulation of a GH millirobot within an isolated porcine bladder (the results of which are now presented in Fig. S12 and Supplementary Video S5). The GH millirobot, guided by a rotating magnetic field of 12 mT, demonstrated its capability to execute targeted movements within the bladder filled with artificial urine. In the first scenario (Fig. S12 I), the GH millirobot navigated from the initial point to target points 1 and 2 sequentially, adhering to a predetermined trajectory designed to maintain distance from the inner wall of the bladder. In the second scenario (Fig. S12 II), the GH millirobot performed multi-angle manipulation around a circular path within the isolated pig bladder, showcasing the advantageous characteristics of its low density, which enabled full suspension and flexible adjustment of movement angles. Throughout these maneuvers, the microrobot effectively maintained a safe distance from the mucosal surface of the bladder wall, thus circumventing potential interference from surface structures—a common challenge encountered in the operation of wall-contact microrobots. Of significant importance, this avoidance of wall contact ensures the protection of non-cancerous cells from exposure to drugs during transport. These demonstrations underscore the advantages of GH millirobots in executing prolonged locomotion and facilitating drug transport within a physiological environment.

Regarding the possibility of fabricating a robot capable of locomoting to the tumor of a mouse while remaining compatible with our imaging system's resolution of 200 um, we appreciate your

suggestion. However, it's essential to consider the limitations imposed by our real-time observation device, Digital Subtraction Angiography (DSA), which operates on X-ray irradiation principles. DSA allows visualization primarily of materials with high density, such as metal, with a resolution of 200 μm for observable materials. Given that our GH millirobots consist predominantly of graphene, with a metal content controlled at a ratio of 6% to maintain suspension characteristics, their visibility under DSA is limited. Consequently, we must adjust the size of the GH millirobots to ensure sufficient metal content for observation. Despite our efforts to experiment with smaller-sized GH millirobots, achieving satisfactory visibility under DSA proved challenging. Therefore, we can only use the rat model to provide locomotion demonstration.

Fig. S12. Time-lapse photographs of the motion of a DOX-HCl@GH millirobot, controlled by a rotating magnetic field of 12 mT, in an isolated porcine bladder. I The GH millirobot was navigated from the initial point to target points 1 and 2 sequentially, adhering to a predetermined trajectory designed to maintain distance from the inner wall of the bladder. **II** The GH millirobot performed multi-angle manipulation around a circular path within the isolated pig bladder, showcasing the advantageous characteristics of its low density, which enabled full suspension and flexible adjustment of movement angles.

Notes: Throughout these maneuvers, the microrobot effectively maintained a safe distance from the mucosal surface of the bladder wall, thus circumventing potential interference from surface structures—a common challenge encountered in the operation of wall-contact microrobots. Of significant importance, this avoidance of wall contact ensures the protection of non-cancerous cells from exposure to drugs during transport. These demonstrations underscore the advantages of GH millirobots in executing prolonged locomotion and facilitating drug transport within a physiological environment.

2) Moreover, if the robot is intended to be used for treating gastric tumours, then what is the purpose of using a swimmer? I would imagine that microswimmers would be more suited for diseases in fluidic environments (i.e. full bladder, eye, blood vessels).

Response:

Thank you for your insightful comment. You raised a good point regarding the suitability of microswimmers for treating gastric tumors, particularly in comparison to diseases in fluidic

environments like the bladder, eye, or blood vessels.

While it's true that microswimmers are generally better suited for diseases within fluidic environments, such as those you mentioned, we need to consider the limitations and practicalities of tumor model construction in other organs. Presently, we do not have suitable technology for establishing tumor models in organs like the bladder, which would be necessary for validating the effectiveness of microswimmers in such environments. Consequently, we opted to demonstrate the feasibility of our approach using gastric tumors as a proof of concept.

Furthermore, it's worth noting that while the stomach may not be a typical fluidic environment, there are instances, such as a time window after the animal has ingested enough water, where the stomach does exhibit fluidic properties and provides ample space for microswimmers to perform their intended tasks effectively.

In order to clarify this point, we have added the following discussion at the end of the in vivo experiment section:

“It's notable that the micro robots demonstrate functionality within the gastric cancer model due to the temporary fluidic environment that the stomach can create following water ingestion. Nonetheless, these micro robots may find enhanced suitability for treating diseases within organs like the bladder, characterized by constant fluidic environments.”

3) Is it possible to observe the tumour and locomote the robot accurately to the target?

Response:

Thanks for the comment. While it is theoretically possible to accurately navigate the robot to the target, we are currently limited by technological constraints that hinder simultaneous execution of (1) magnetic field navigation of the microrobot and (2) real-time visualization of the tumor. Consequently, we conducted ex vivo manipulation of a GH millirobot in an isolated porcine bladder. In this context, the GH millirobot demonstrated precise movement to the designated target point (tumor). The experimental process can be found in the supplementary video S5.

4) Is the drug carried by a single helical swimmer sufficient? And if it is not, then are there strategies to overcome the insufficient drug dose, considering that the authors already ruled out swarm control?

Response:

Thank you for your question. Indeed, the drug carried by a single helical swimmer may be insufficient. To address this limitation, we can increase the number of swimmers carrying the drug. While these swimmers may not exhibit deformation akin to swarm robotics, we can effectively control a large number of helical swimmers to move in unison towards the target, thereby overcoming the challenge of insufficient drug dosage.

Having said that, I think that the authors did satisfactorily address Reviewer 2's comments.

Response: Thank you for acknowledging our efforts in addressing the reviewers' comments and for your support of our work.

REVIEWERS' COMMENTS

Reviewer #3 (Remarks to the Author):

The authors addressed all the comments, included necessary clarifications and performed appropriate language corrections.

Reviewer #4 (Remarks to the Author):

The authors have addressed my comments.